# Rethinking Heavy Models in Multivariate Time Series Anomaly Detection

## Abstract

Multivariate time series anomaly detection (MTS-AD) is widely used, but real-world deployments often face tight computational budgets that limit the practicality of deep learning. We revisit whether heavy deep models (high-FLOPs architectures) are necessary to achieve strong detection performance in such settings. We conduct a systematic, compute-aware comparison of statistical, classical machine learning, and deep learning methods across diverse MTS-AD benchmarks, measuring detection with AUROC (threshold-free, thus application-agnostic) and cost with FLOPs (a hardware-agnostic proxy enabling fair cross-method comparison). We find that traditional approaches often match or surpass deep models, which appear less frequently among the top performers, and that the effectiveness-efficiency trade-off commonly favors non-deep alternatives under limited budgets. These results indicate that deep learning is not uniformly superior for MTS-AD and that heavy architectures can be counterproductive in resource-constrained deployments. These findings offer practical guidance for practitioners designing anomaly monitoring systems under compute constraints, highlighting cases where lightweight models are sufficient and heavy deep models may be worth the cost.

## 1 Introduction

Time series anomaly detection is a fundamental task in machine learning with wide-ranging applications in domains such as industrial control systems, aerospace telemetry, and cyber security (Kim et al., 2023; Hundman et al., 2018; Landauer et al., 2025). In practice, anomalies are rare and difficult to label, which makes unsupervised anomaly detection methods trained on normal data an essential approach. Over the past decade, deep learning methods have gained prominence for anomaly detection, achieving impressive performance across a variety of benchmark datasets (Zamanzadeh Darban et al., 2024).

However, real-world deployment environments often impose severe hardware and operational constraints. For example, safety-critical systems may need to operate without external connectivity due to security restrictions, preventing the use of cloud-based solutions (Bhamare et al., 2020). Similarly, embedded monitoring devices may lack GPUs or operate under strict thermal and power limitations, making it impractical to deploy computationally intensive deep learning methods (Shuvo et al., 2023; Singh & Gill, 2023). In such cases, the assumption that deep learning is the universally superior solution becomes questionable. While recent research has emphasized novel neural architectures, comparatively little work has jointly examined both effectiveness and efficiency under constrained computing conditions. Most studies focus on accuracy alone (Jia et al., 2025), with only a few recent benchmarks considering accuracy together with runtime and memory usage (Qiu et al., 2025). In addition, several works have highlighted inconsistencies in evaluation protocols and metrics for time series anomaly detection in industry (Si et al., 2024).

This gap encourages us to consider two central questions: (i) What are the most effective options for time series anomaly detection under limited computational resources, and are deep learning methods always the best options? (ii) Does a trade-off between detection performance and computational cost truly exist in practice? Our own experience in industrial applications, including monitoring of air defense systems and equipment in manufacturing settings, has made clear the difficulty of balancing computational demands with detection performance. We believe that many practitioners working in real-world deployments encounter the same challenge.

Table 1: Summary of resource-constrained environments

| Typical Domain | Representative Constraint | Common Approach | Key References |
|---|---|---|---|
| **Limited memory** | | | |
| IoT/Wireless Sensor Network nodes; streaming telemetry in manufacturing and equipment monitoring | devices cannot buffer long histories; models must adapt with small working sets under drift | online/streaming learning; feature selection; memory-efficient summaries/sketches | Bifet & Gavaldà (2007) Nancy et al. (2020) Jain et al. (2022) Chatterjee & Ahmed (2022) Mfondoum et al. (2024) |
| **GPU unavailable** | | | |
| ICS and OT; manufacturing cells; safety/certification-constrained environments | power/thermal, enclosure, and certification constraints preclude accelerators; inference must be CPU-only on-prem | quantization; pruning/compression; CPU-optimized runtime | Han et al. (2016) Jacob et al. (2018) Sipola et al. (2022) Das & Luo (2023) Singh & Gill (2023) Liu et al. (2024a) Fährmann et al. (2025) |
| **Limited CPU capacity** | | | |
| PLC/RTU-adjacent controllers; fanless industrial PCs; battery-powered sensing | very limited CPU cycles and RAM; strict cycle-time determinism | quantization; low-FLOPs model design; online/streaming updates; selective features | Liu et al. (2008) Goldstein & Dengel (2012) Singh & Gill (2023) |
| **Restricted communication** | | | |
| air-gapped ICS/OT; secure manufacturing cells; remote or intermittently connected sites | on-prem/offline operation and stringent latency disallow cloud round-trips; data egress may be restricted | local inference at the edge; federated/on-site adaptation; minimal upstream telemetry | Belenguer et al. (2022) Das & Luo (2023) Dehlaghi-Ghadim et al. (2023) Stouffer et al. (2023) |

In this paper, we therefore address the questions by conducting a systematic comparative study of unsupervised anomaly detection methods that range from traditional approaches to deep learning methods. Unlike prior studies that have primarily emphasized accuracy, we introduce an evaluation framework that considers both detection performance and computational cost (Mejri et al., 2024). This perspective enables a fair comparison across different methodological approaches. Our evaluation covers diverse real-world datasets drawn from industrial, server, and aerospace domains, ensuring that our findings generalize across multiple application settings.

## 2 LITERATURE REVIEW

### 2.1 RESOURCE CONSTRAINED ENVIRONMENTS

The deployment of models in industrial system is not solely governed by algorithmic accuracy but is equally constrained by system-level limitations. As summarized in Table 1, the literature consistently highlights four recurring scenarios in resource-constrained environments, which encompass limited memory, GPU unavailable, limited CPU capacity, and restricted communication. These scenarios illustrate the historical progression of research toward resource-aware solutions and motivate the comparative analysis conducted in this study.

**Limited memory** Several studies have demonstrated that real-world industrial environments, including IoT nodes, wireless sensor networks, and manufacturing telemetry systems, often operate under severe limitations in storage and energy, making the buffering of long historical windows infeasible (Jain et al., 2022; Mfondoum et al., 2024; Chatterjee & Ahmed, 2022). In time series anomaly detection, such constraints require models to process data incrementally while maintaining only a limited working set. Consequently, prior research has emphasized memory-efficient representations, dimensionality reduction through feature selection (Nancy et al., 2020), and online or adaptive windowing techniques, which dynamically adjust to evolving conditions (Bifet & Gavaldà, 2007).

**GPU unavailable** In operational technology (OT) and industrial control system (ICS) environments, the use of GPU accelerators is often infeasible due to strict power, thermal, and certification constraints (Das & Luo, 2023; Liu et al., 2024a; Singh & Gill, 2023; Sipola et al., 2022). As a result, inference is typically performed on CPUs, making model-level optimization essential. In

this context, two advances stand out as particularly effective. Integer-only quantization executes inference entirely with INT8 arithmetic, while deep compression combines pruning and quantization to reduce both computation and memory requirements. These techniques together enable efficient CPU-centric optimization (Fährmann et al., 2025; Jacob et al., 2018; Han et al., 2016).

**Limited CPU capacity** A related constraint emerges when devices rely on modest CPUs, such as PLC/RTU-adjacent controllers, fanless industrial PCs, or battery-powered IoT nodes. In these settings, the need for cycle time determinism combined with limited computational throughput requires fundamentally low complexity designs. Traditional detectors such as Isolation Forest or histogram-based method remain attractive due to their favorable time and memory complexity (Liu et al., 2008; Goldstein & Dengel, 2012). Recent surveys on Edge AI further emphasize that such CPU-constrained deployments require models explicitly tailored to minimize FLOPs while preserving detection capability (Singh & Gill, 2023; Sipola et al., 2022).

**Restricted communication** In many industrial and critical infrastructure domains, data transfer to the cloud is either infeasible or prohibited due to latency requirements and strict security policies. Authoritative guideline from National Institute of Standard and Technology explicitly recommend isolation of OT networks, reinforcing the necessity of on-device or on-premise models (Stouffer et al., 2023). Research has therefore explored federated learning approaches to enable collaborative learning without raw data sharing, particularly in distributed industrial environments, as well as lightweight edge frameworks (Belenguer et al., 2022; Dehlaghi-Ghadim et al., 2023).

## 2.2 TIME SERIES ANOMALY DETECTION ALGORITHMS

Early research on time series anomaly detection was largely grounded in basic statistical models, which assumed stationary distributions and relatively simple dependency structures. Among the most influential were multivariate monitoring methods such as Hotelling's $T^2$ statistic that leveraged distributional thresholds to detect deviations in industrial manufacturing processes (H.Hotelling, 1947; Ye & Chen, 2001; Zheng et al., 2016). However, their reliance on linearity and stationarity assumptions rendered them less effective when confronted with the high-dimensional, noisy, and non-stationary signals that characterize modern industrial systems.

The subsequent wave of research introduced non-parametric machine learning approaches that relaxed restrictive distributional assumptions. Distance and density-based detectors identified anomalies as local deviations within the data manifold (Angiulli & Pizzuti, 2002; Breunig et al., 2000), while clustering-based techniques grouped time series patterns to distinguish normal from abnormal behavior (He et al., 2003). Ensemble-based strategies, such as Isolation Forest, improved robustness and scalability through randomized partitioning and aggregation (Liu et al., 2008).

Driven by advances in representation learning, the field has recently shifted toward deep learning approaches that explicitly model sequential dependencies and nonlinear structures. Early works employed recurrent neural networks and detected anomalies by reconstructing temporal sequences (Malhotra et al., 2016; Park et al., 2018). This paradigm was later extended by probabilistic generative models, adversarially trained architectures, and attention-based models (Su et al., 2019; Li et al., 2019; Geiger et al., 2020; Zhou et al., 2019; Akcay et al., 2019; Tuli et al., 2022; Xu et al., 2022; Wu et al., 2023). Collectively, these approaches represent a clear trajectory toward increasingly complex and expressive models, often achieving state-of-the-art accuracy across widely used benchmarks. Nevertheless, their heavy reliance on GPU accelerators and large memory footprints has raised practical concerns regarding deployment in resource-constrained industrial environments.

To contextualize the performance of these approaches, several benchmark studies have systematically compared classical and deep learning methods (Han et al., 2022; Paparrizos et al., 2022; Si et al., 2024). However, most prior studies emphasize accuracy while giving limited attention to computational cost. In this work, we provide a more balanced evaluation by jointly examining detection performance and computational cost across both traditional and deep learning models. Further discussion of the scope and limitations for existing benchmarks is presented in the Appendix A.

## 3 EXPERIMENTAL DESIGN

### 3.1 PROBLEM DEFINITION

#### 3.1.1 UNSUPERVISED ANOMALY DETECTION IN TIME SERIES

Unsupervised anomaly detection in multivariate time series is the task of identifying abnormal behaviors without access to anomaly labels during training (Pang et al., 2021). This setting is common in practice because anomalies are rare, labeling is costly (Blázquez-García et al., 2021).

A multivariate time series is defined as $x \in \mathbb{R}^{N \times D}$ and can be expressed as

$$X = \{x_t \in \mathbb{R}^D \mid t = 1, \ldots, N\} \tag{1}$$

where $N$ denotes the sequence length and $x_t$ is the $D$-dimensional observation at time $t$. Based on the learned representation of normal behaviors, the model provides a decision function that assigns each observation an anomaly score

$$s_t = F(x_t), \quad \{s_t\}_{t=1}^N \in \mathbb{R}^N \tag{2}$$

where observations with higher scores are regarded as anomalies, typically determined by calibrating a threshold from training scores (Su et al., 2019; Audibert et al., 2020; Xu et al., 2022).

The unsupervised formulation is particularly important in real-world applications, since annotated anomalies are typically unavailable, occur infrequently, or vary significantly across domains (Salehi & Rashidi, 2018). By modeling normality directly from unlabeled data, unsupervised approaches provide a practical and general framework for anomaly detection in time series.

#### 3.1.2 RESEARCH QUESTIONS

Building on the above definition, we aim to investigate how unsupervised anomaly detection in multivariate time series can be evaluated not only in terms of effectiveness but also efficiency under realistic deployment settings. While prior work has primarily emphasized improving detection accuracy, comparatively less attention has been paid to the computational and operational feasibility of different methods. To address this gap, we define the following research questions:

**(i) What are the most effective options for time series anomaly detection under limited computational resources, and are deep learning methods always the best options?** This question is motivated by the observation that many deployment environments face practical constraints, including restricted computational capacity, memory, or energy availability. Although deep learning methods have shown strong benchmark performance, their dependence on substantial resources raises doubts about their universal applicability. It is therefore important to examine whether traditional statistical or machine learning methods may provide more practical alternatives under such constrained conditions.

**(ii) Does a trade-off between detection performance and computational cost truly exist in practice?** Deep learning models are generally associated with higher computational cost due to their larger architectures and resource requirements. The literature also shows that traditional statistical and machine learning methods can remain competitive in certain scenarios. This raises the question of whether such cost is justified by consistently superior detection performance. This question therefore seeks to clarify whether higher computational cost truly translates into superior performance, or whether certain approaches can offer a more balanced relationship that challenges the prevailing view.

### 3.2 EVALUATION PROTOCOL

#### 3.2.1 CHOICE OF METHODS AND DATASETS

**Algorithms** We survey the historical development of unsupervised time series anomaly detection and curate a representative set of models for evaluation. The selection of models follow three principles. First, to ensure chronological coverage, we consider landmark contributions from the early multivariate statistical monitoring methods through contemporary deep learning methods, so

that each major period is represented. Second, to reflect differences in operating mechanisms, we partition the candidate methods into three families, namely statistical, one-class classification, and reconstruction-based, and we select representative exemplars from each family. Third, we focus on models that are well-established in the field and have been widely adopted in prior studies. Detailed descriptions of the included methods are provided in the Appendix B.1.

**Datasets**  To support reproducible and deployment relevant evaluation, we prioritize datasets that capture real operational complexity and are widely used in the literature. We also focus on datasets from machinery and electronic system contexts where anomaly detection techniques are routinely deployed. Accordingly, we include SMD (Su et al., 2019) and PSM (Abdulaal et al., 2021) from server environments, SMAP and MSL from NASA engineering telemetry (Hundman et al., 2018), and SWaT (Mathur & Tippenhauer, 2016) and WADI (Ahmed et al., 2017) from industrial water treatment and distribution testbeds. Each dataset is multivariate and displays cross-channel dependencies and non-stationary dynamics, with rare anomalies as in real-world environments. Detailed descriptions of the included datasets are provided in the Appendix B.2.

### 3.2.2 METRICS

**Detection Performance**  We compare detection performance across models using a threshold-agnostic metric, the Area Under the Receiver Operating Characteristic Curve (AUROC), to ensure fairness. Although certain methods are often paired with post hoc thresholding procedures such as Peaks Over Threshold, we do not apply such schemes to evaluate the intrinsic ranking quality of each model. To ensure a uniform basis of comparison, each implementation outputs a real-valued anomaly score at every time step, aligned with the original sequence length. AUROC is therefore computed at the same temporal resolution for all methods.

**Computational Cost**  To enable a fair comparison of computational demands across both traditional algorithms and deep learning models, we adopt floating-point operations (FLOPs) as a unifying metric. FLOPs are model-agnostic and can be meaningfully related to hardware capabilities, making them particularly suitable for analyzing hardware-constrained scenarios. Unlike elapsed real time, FLOPs isolate algorithmic complexity from hardware variability. However, estimating FLOPs is nontrivial because the operations that dominate computational cost differ substantially across models and are highly sensitive to hyperparameter settings (e.g. tree depth in ensemble methods, number of neighbors in $k$-NN, or hidden dimension in neural networks). Consequently, prior work has rarely reported FLOPs for traditional machine learning methods, focusing only on deep learning models. We therefore derive closed-form or tight counting formulas for each traditional method and instantiate them with the exact hyperparameters used in our experiments. For deep learning models, we employed the PyTorch-based package `calflops`, which provides automatic FLOPs accounting given model architectures and input shapes (Ye, 2023).

In conducting FLOPs estimation, we adhere to the following principles:

- **Dataset specificity** FLOPs are computed separately for each dataset, as input dimensionality, sequence length, and sample size directly affect operation counts.
- **Training vs. inference** We compute FLOPs for both training and inference phases, reflecting their distinct computational characteristics.
- **Epoch sensitivity in deep learning models** Because training until convergence is ambiguous and dependent on hyperparameter optimization, we report FLOPs for a single epoch of training and the epoch at which the best AUROC is achieved during hyperparameter search.
- **Fair treatment of comparison operations** While FLOPs conventionally account only for addition and multiplication, algorithms such as $k$-NN and distance-based methods (ABOD, LOF) are dominated by comparison operations. Excluding these will unfairly understate their complexity, therefore, we count each comparison as a single FLOP.
- **Data-dependent structures** For models where computational complexity depends on data distribution such as clustering-based models, FLOPs are computed from the fitted model structure on the actual data rather than theoretical worst-case bounds.
- **Approximation for non-primitive operations** For procedures such as sorting, where exact FLOPs are impractical to enumerate, we adopt widely accepted complexity-based approximations (e.g. $O(n \log n)$, where $n$ is the number of instances).

Table 2: Operation counts (FLOPs) for each method. We count additions, multiplications, and comparisons equally as 1 FLOP. The notation $n_{\text{tr}}$, $n_{\text{inf}}$, and $d$ denote the number of training instances, the number of test instances, and the input dimension, respectively. Model-specific notations are explained in the Notation column.

| Model | Type | FLOPs | Notation |
|---|---|---|---|
| Hotelling | Train | $2n_{\text{tr}}d^2 + 2n_{\text{tr}}d + d^3$ | |
| | Inference | $n_{\text{inf}}(2d^2 + 2d - 1)$ | |
| PCA | Train | $2n_{\text{tr}}d^2 + 2n_{\text{tr}}d + 3d^2$ | • $p$: # of PCA components |
| | Inference | $n_{\text{inf}}(4pd - p + 2d - 1)$ | |
| ABOD | Train | $1.5n_{\text{tr}}(n_{\text{tr}} - 1)d$ $+n_{\text{tr}}(n_{\text{tr}} - 1)\log_2(n_{\text{tr}} - 1)$ $+n_{\text{tr}}k(k - 1)(d + 2) + n_{\text{tr}}$ | • $k$: # of neighbors |
| | Inference | $1.5n_{\text{inf}}(n_{\text{inf}} - 1)d$ $+n_{\text{inf}}(n_{\text{inf}} - 1)\log_2(n_{\text{inf}} - 1)$ $+n_{\text{inf}}k(k - 1)(d + 2) + n_{\text{inf}}$ | |
| LOF | Train | $1.5n_{\text{tr}}(n_{\text{tr}} - 1)d$ $+n_{\text{tr}}(n_{\text{tr}} - 1)\log_2(n_{\text{tr}} - 1)$ $+n_{\text{tr}}k + n_{\text{tr}}(k + 1) + 2n_{\text{tr}}k$ | • $k$: # of neighbors |
| | Inference | $1.5n_{\text{inf}}(n_{\text{inf}} - 1)d$ $+n_{\text{inf}}(n_{\text{inf}} - 1)\log_2(n_{\text{inf}} - 1)$ $+n_{\text{inf}}k + n_{\text{inf}}(k + 1) + 2n_{\text{inf}}k$ | |
| CBLOF | Train | $n_{\text{tr}}I(3Cd + d - 1) + 3d((n_{\text{tr}} - |LC|)L + |LC|)$ | • $I$: max iterations for clustering 
 • $C$: # of clusters 
 • $L$: # of large clusters 
 • $|LC|$: # of instances in large clusters |
| | Inference | $n_{\text{inf}}I(3Cd + d - 1) + 3d((n_{\text{inf}} - |LC|)L + |LC|)$ | |
| HBOS | Train | $2n_{\text{tr}}d + 5bd$ | • $b$: # of bins |
| | Inference | $3n_{\text{inf}}d + 2bd$ | |
| LODA | Train | $n_{\text{tr}}c(2\sqrt{d} + \log_2 b - 1)$ | • $b$: # of bins 
 • $c$: # of random cuts |
| | Inference | $n_{\text{inf}}(2c\sqrt{d} + c\log_2 b + 1)$ | |
| Isolation Forest | Train | $T(2s\log_2 s)$ | • $T$: # of estimators 
 • $s$: max samples per estimator 
 • $\gamma$: Euler–Mascheroni const. |
| | Inference | $n_{\text{inf}}(T(2\log s - 2 + \gamma) - 2(1 - 1/s) + (T + 2))$ | |
| HS-Tree | Train | $T(\psi(h + 1) + 5(2^{h+1} - 1))$ | • $T$: # of estimators 
 • $h$: max depth of tree 
 • $\psi$: reference window size |
| | Inference | $n_{\text{inf}}T(5h + 7)$ | |

We summarize the final FLOPs formulations of the chosen methods in Table 2 and present their detailed derivations in Appendix C.

## 4 RESULTS AND ANALYSES

In this section we present three experiments to examine performance versus computational cost, estimated time comparison under hardware-constrained settings, and model scalability. The first experiment evaluates the trade-off between detection performance, mainly measured by AUROC, and the computational cost quantified by training and inference FLOPs. For traditional models, training and inference FLOPs are given in Table 2. For deep learning models, we report training FLOPs per epoch, and full-training FLOPs correspond to the sum overall epochs. In subsequent analyses, we use full-training FLOPs for deep learning models. The second experiment leverages the fact that, compared to other efficiency metrics, FLOPs have the key advantage of being directly comparable to hardware performance, typically expressed as floating-point operations per second (FLOPS),

Table 3: Top five models for each dataset ranked in descending order of AUROC. All computational costs are reported in gigaFLOPs (GFLOPs). AUROC and AUPRC scores are reported as the mean over five runs with different random seeds.

| Dataset | Model | Type | GFLOPs ↓ | | | AUROC ↑ | AUPRC ↑ |
| | | | Train | Inference | Full-training | | |
|---|---|---|---|---|---|---|---|
| PSM | Hotelling | Statistical | 0.17 | 0.11 | — | $0.77 \pm 0.00$ | $0.49 \pm 0.00$ |
| | LSTM-AE | Reconstruction | 12.23 | 2.76 | 1223.5 | $0.76 \pm 0.01$ | $0.50 \pm 0.04$ |
| | ABOD | Statistical | 926.33 | 421.87 | — | $0.75 \pm 0.00$ | $0.43 \pm 0.00$ |
| | LOF | One-class | 917.76 | 416.07 | — | $0.73 \pm 0.00$ | $0.42 \pm 0.00$ |
| | HBOS | Statistical | 0.01 | 0.01 | — | $0.73 \pm 0.00$ | $0.50 \pm 0.00$ |
| MSL | CBLOF | One-class | 0.89 | 2.60 | — | $0.65 \pm 0.01$ | $0.20 \pm 0.01$ |
| | HS-Tree | One-class | $< 0.01$ | 0.08 | — | $0.64 \pm 0.03$ | $0.14 \pm 0.03$ |
| | TimesNet | Reconstruction | 21851.23 | 9208.69 | 437024.6 | $0.64 \pm 0.03$ | $0.16 \pm 0.01$ |
| | ABOD | Statistical | 334.71 | 536.74 | — | $0.63 \pm 0.00$ | $0.17 \pm 0.00$ |
| | HBOS | Statistical | 0.01 | 0.01 | — | $0.62 \pm 0.00$ | $0.16 \pm 0.00$ |
| SMAP | Isolation Forest | One-class | $< 0.01$ | 0.48 | — | $0.64 \pm 0.01$ | $0.16 \pm 0.00$ |
| | ABOD | Statistical | 998.15 | 10281.98 | — | $0.64 \pm 0.00$ | $0.17 \pm 0.00$ |
| | LOF | One-class | 996.78 | 10277.64 | — | $0.62 \pm 0.00$ | $0.17 \pm 0.00$ |
| | CBLOF | One-class | 2.48 | 10.76 | — | $0.62 \pm 0.01$ | $0.16 \pm 0.00$ |
| | HBOS | Statistical | 0.01 | 0.03 | — | $0.61 \pm 0.00$ | $0.15 \pm 0.00$ |
| SMD | TimesNet | Reconstruction | 25325.13 | 8441.89 | 506502.64 | $0.77 \pm 0.00$ | $0.17 \pm 0.00$ |
| | LSTM-AE | Reconstruction | 82.70 | 27.57 | 8269.64 | $0.77 \pm 0.01$ | $0.18 \pm 0.01$ |
| | Hotelling | Statistical | 2.10 | 2.10 | — | $0.73 \pm 0.00$ | $0.16 \pm 0.00$ |
| | CBLOF | One-class | 36.30 | 34.65 | — | $0.72 \pm 0.01$ | $0.16 \pm 0.01$ |
| | ABOD | Statistical | 38426.94 | 38428.58 | — | $0.71 \pm 0.00$ | $0.10 \pm 0.00$ |
| SWaT | HBOS | Statistical | 0.05 | 0.07 | — | $0.85 \pm 0.00$ | $0.75 \pm 0.00$ |
| | Isolation Forest | One-class | $< 0.01$ | 0.36 | — | $0.83 \pm 0.00$ | $0.73 \pm 0.01$ |
| | OmniAnomaly | Reconstruction | 51.30 | 15.49 | 1026.05 | $0.83 \pm 0.00$ | $0.73 \pm 0.00$ |
| | LODA | Statistical | 1.39 | 1.39 | — | $0.82 \pm 0.02$ | $0.73 \pm 0.02$ |
| | PCA | Reconstruction | 2.64 | 2.42 | — | $0.82 \pm 0.00$ | $0.73 \pm 0.00$ |
| WADI | HBOS | Statistical | 0.19 | 0.06 | — | $0.74 \pm 0.00$ | $0.18 \pm 0.00$ |
| | Isolation Forest | One-class | $< 0.01$ | 0.33 | — | $0.74 \pm 0.02$ | $0.18 \pm 0.01$ |
| | LODA | Statistical | 1.46 | 0.34 | — | $0.72 \pm 0.04$ | $0.27 \pm 0.05$ |
| | TimesNet | Reconstruction | 24899.00 | 1828.07 | 497979.94 | $0.66 \pm 0.01$ | $0.20 \pm 0.00$ |
| | HS-Tree | One-class | $< 0.01$ | 0.20 | — | $0.63 \pm 0.05$ | $0.09 \pm 0.02$ |

where FLOPs denote algorithmic operation counts and FLOPS denote hardware throughput. Dividing algorithmic FLOPs by the FLOPS of a target hardware allows us to approximate training and inference time under GPU-free, hardware-constrained deployments. The third experiment examines scalability by varying the number of instances and feature dimensions, thereby observing how computational cost changes with dataset size. Model hyperparameters are chosen as those yielding the highest AUROC within the defined search space, with the detailed specification of the search space given in Appendix D.

## 4.1 Performance vs. Efficiency

Table 3 summarizes the main results, listing the top five models ranked by AUROC and allowing a direct inspection of which methods simultaneously offer high detection performance and favorable computational properties. Across individual datasets, high AUROC values are frequently achieved by approaches with relatively low computational cost. Although deep learning models often appear among the top performers in certain cases, their advantages are neither consistent nor substantial across datasets.

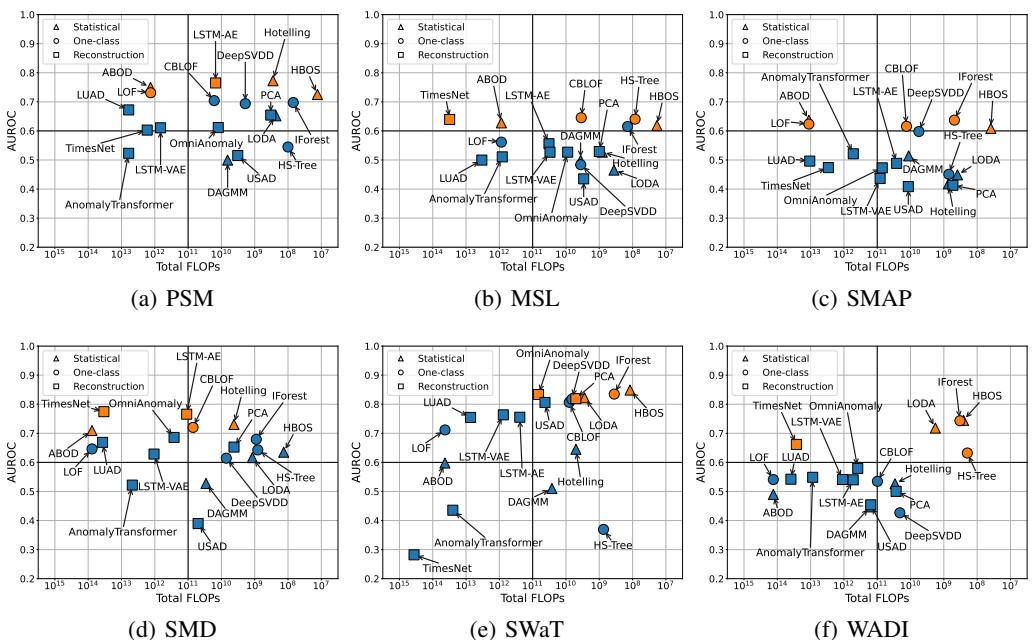

Figure 1: AUROC vs. Total FLOPs (sum of training and inference FLOPs) for each dataset. Orange markers denote the top five models in terms of AUROC.

Figure 1 further illustrates this trend by plotting AUROC against the total FLOPs, obtained by summing training and inference FLOPs, with the top-performing models highlighted in orange. A closer examination of the highlighted points shows that many traditional methods, aside from the $k$-NN family including ABOD and LOF, tend to attain high AUROC while remaining concentrated in the low-FLOPs region. These models therefore constitute practical and preferable alternatives in settings where computational resources are limited. It is also evident that approaches relying on $k$-NN exhibit noticeably higher FLOPs due to their quadratic complexity, which restricts their applicability in large-scale or resource-constrained environments.

In scenarios where computational resources are severely limited, the practical priority often shifts from maximizing raw detection accuracy to minimizing computational cost. To reflect this consideration, an additional table highlighting the five most computationally efficient models in terms of FLOPs is presented in Appendix E.1. This FLOPs-oriented view is intended for deployments where computational efficiency is essential and therefore offers a complementary perspective to the main results. We also report results for AUPRC, VUS-ROC, and VUS-PR. These metrics exhibit trends that closely align with AUROC, reinforcing the overall conclusion that models with low computational overhead generally achieve a favorable balance between detection performance and operational feasibility. Entire results for all models and datasets are also included in Appendix E.1.

## 4.2 COMPARISON OF ESTIMATED TIME

Estimate of the minimal execution time can be calculated as FLOPs/FLOPS. To capture variability across deployment conditions, we assume three hardware scenarios: a highly resourced environment represented by our experimental setup (Intel(R) Core(TM) i9-14900K CPU, NVIDIA GeForce RTX 5070 Ti GPU), a mobile environment corresponding to Samsung Galaxy A32 (Cortex-A75 & Cortex-A55 CPU, Arm Mali-G52 MC2 GPU), and a resource-constrained edge environment represented by Raspberry Pi 3B+ (Asutkar et al., 2023; Trilles et al., 2024). The FLOPS of each device are computed using only the number of cores $C$ and clock frequency $f$, as formalized in Equation 3.

$$\text{FLOPS}_{CPU}^{peak} = \sum_{t \in \{\text{Performance,Efficient}\}} C_t \times f_t, \quad \text{FLOPS}_{GPU}^{peak} = C \times f \tag{3}$$

A practically significant observation emerging from Figure 2 is that deep learning models are feasible only in highly resourced environments, where the abundance of GPU cores substantially mit-

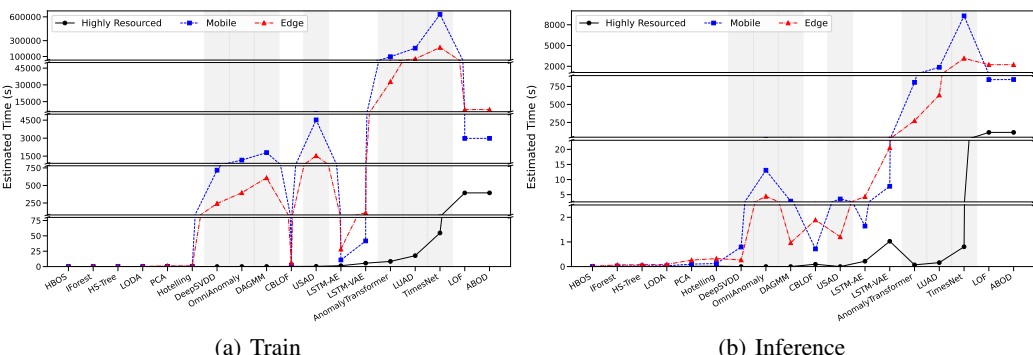

|  (a) Train | (b) Inference |

Figure 2: Estimated execution time of anomaly detection models under three hardware scenarios (Highly Resourced, Mobile, and Edge). Models are ordered in ascending runtime according to the Highly Resourced scenario, with deep learning models highlighted by gray shading. The results are averaged across datasets.

igates their computational burden. In sharp contrast, under mobile or edge scenarios these models incur prohibitively high costs, making their deployment virtually infeasible. By comparison, tree-based algorithms such as Isolation Forest and HS-Tree, along with histogram-based algorithms such as HBOS and LODA, consistently maintain extremely low computational overhead during both training and inference across all hardware settings. Finally, approaches that rely on $k$-NN, including ABOD and LOF, exhibit comparatively elevated costs, underscoring their limited practicality in data-abundant contexts. Results for individual datasets are provided in Appendix E.2.

To strengthen the practical relevance of these findings, we additionally report real runtime measurements under our highly resourced environment while disabling GPU acceleration to ensure a fair, CPU-only comparison. Using the SMD dataset and selecting representative models from each methodological category, HBOS required 3.27 seconds for training and 0.77 seconds for inference, while Isolation Forest required 4.97 seconds and 2.01 seconds, respectively. In contrast, the deep learning model USAD incurred 1047.6 seconds for training and 5.45 seconds for inference under the same setting. Finally, we also evaluated ABOD, which recorded the highest estimated time among all methods, and observed prohibitively high costs of 4700.05 seconds for training and 6331.64 seconds for inference. These results align closely with the FLOPs based trends and further substantiate the practical infeasibility of heavy models in resource-constrained deployments.

## 4.3 MODEL SCALABILITY

As illustrated in Figure 3, tree-based methods such as Isolation Forest and HS-Tree demonstrate high scalability, with computational requirements scaling sublinearly with respect to $n$, thereby making them well-suited for large-scale deployment. In contrast, methods using $k$-NN such as ABOD and LOF, exhibit a steep rise in both training and inference FLOPs due to pairwise distance computations, making them impractical for large datasets. Deep learning models exhibit a consistently linear increase in FLOPs with data size. Although they scale more favorably than $k$-NN approaches, their computational cost still poses a substantial burden as $n$ becomes large.

A similar pattern appears when varying the feature dimension. Tree-based and histogram-based approaches, such as HS-Tree, Isolation Forest and HBOS, show only less sensitivity to increasing dimensionality, maintaining relatively low FLOPs across the range of dimensions. By contrast, $k$-NN methods increase linearly with dimension because each distance computation scales directly with the number of features. Deep learning models also scale approximately linearly, while their overall computational demand remains high. A particularly distinctive trend in dimensional scalability test is that Hotelling's $T^2$ grows even more rapidly as dimension increases, since estimating and inverting the covariance matrix requires $O(d^2)$ and $O(d^3)$ operations, respectively.

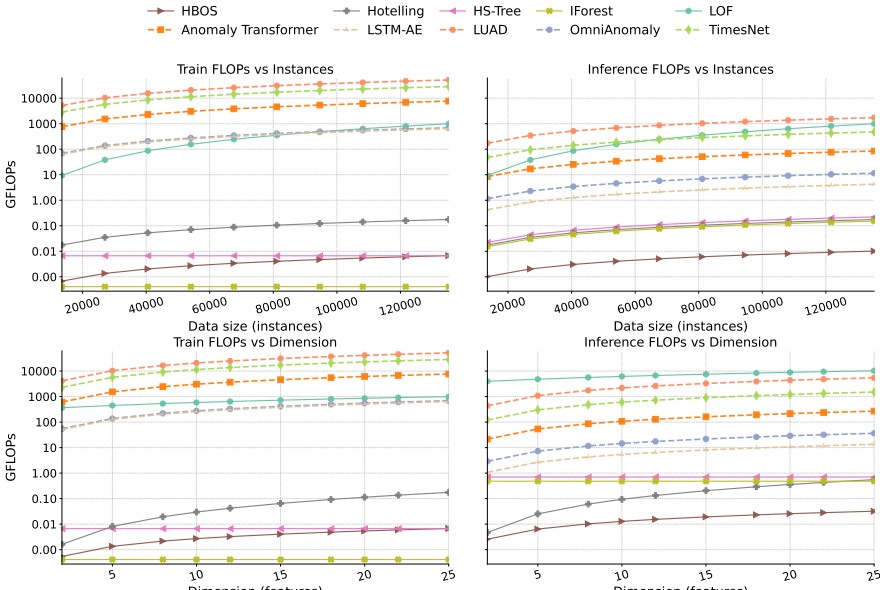

Figure 3: Training (left) and inference (right) FLOPs of ten representative models on the SMAP dataset, plotted against the number of instances (top) and feature dimension (bottom). Both axes for FLOPs use a logarithmic scale.

This divergence underscores the importance of considering algorithmic scalability when selecting anomaly detection models for real-world applications where data volumes can be substantial. Entire results for all models and datasets are provided in Appendix E.3.

## 5 CONCLUSION

This work presented a systematic comparison of traditional and deep learning methods for unsupervised time series anomaly detection under resource-constrained settings. By jointly evaluating AUROC and FLOPs, we explored two central questions concerning the effectiveness of models under limited resources and whether and how performance differences arise between traditional and deep learning methods. Our results showed that traditional models often rank among the top performers, deep learning models tend to do not consistently surpass them, and the balance between effectiveness and efficiency favors traditional approaches except for $k$-NN methods. Estimated time analysis further revealed that while deep learning models are feasible only in highly resourced environments, traditional models remain practical under resource-constrained settings while offering comparable detection performance. Overall, these findings challenge the view that deep learning is always the superior choice and emphasize the continued viability of traditional methods for real-world deployment. In doing so, we aim for this work to inspire new avenues of research while also providing practitioners with a useful point of reference when building anomaly detection systems under real-world constraints.

### REPRODUCIBILITY STATEMENT

We have undertaken several efforts to ensure the reproducibility of our work. FLOPs derivations for all algorithms are provided in Appendix C, and the definition of hyperparameter search spaces is given in Appendix D. Extensive experimental results across datasets are presented in Appendix E. The supplementary material contains our full experimental pipeline, including model implementations, configuration files, and data preprocessing scripts, thereby facilitating independent reproduction of our findings.

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

# A    PREVIOUS BENCHMARKS

Table 4: Previous benchmark papers considering computational cost.

| Paper Type | Paper | Cost Metric | Coverage Model Type | | | Data Source | Dimension |
|---|---|---|---|---|---|---|---|
| | | | Stat. | ML | DL | Real | Multi |
| Benchmark | Schmidl et al. (2022) | Memory, Time | ✓ | ✓ | ✓ | ✓ | ✓ |
| | Paparrizos et al. (2022) | Time | ✓ | ✓ | ✓ | ✓ | ✗ |
| | Han et al. (2022) | Time | ✓ | ✓ | ✓ | ✓ | ✓ |
| | Dobos et al. (2023) | Time | ✓ | ✓ | ✓ | ✗ | ✓ |
| | Rewicki et al. (2023) | Time | ✓ | ✓ | ✓ | ✓ | ✗ |
| | Liu & Paparrizos (2024) | Time | ✓ | ✓ | ✓ | ✓ | ✓ |
| | Si et al. (2024) | Time | ✓ | ✗ | ✓ | ✓ | ✓ |
| | Qiu et al. (2025) | Memory, Time | ✓ | ✓ | ✓ | ✗ | ✓ |
| Methodology | Xu et al. (2023) | Params, FLOPs, Time | ✗ | ✗ | ✓ | ✓ | ✓ |
| | Liu et al. (2024b) | Params, FLOPs, Time | ✗ | ✗ | ✓ | ✓ | ✓ |
| | Ho & Armanfard (2025) | FLOPs, Time | ✗ | ✗ | ✓ | ✓ | ✓ |
| **Ours** | | **FLOPs** | ✓ | ✓ | ✓ | ✓ | ✓ |

As summarized in Table 4, a number of benchmark studies on multivariate time series anomaly detection have undertaken extensive comparisons across models and datasets. Despite this broad of coverage, the evaluation of computational cost in these works remains limited in scope. Most benchmarks have relied primarily on execution time as the cost metric, with a few additionally considering memory usage. However, runtime measurements are inherently dependent on hardware specifications and experimental settings, which constrains their comparability across studies. Likewise, memory consumption does not fully capture the algorithmic complexity of the models and therefore provides only a partial view of computational efficiency.

Beyond benchmark papers, some methodology studies have employed hardware-agnostic measures such as parameter counts or FLOPs. However, these comparisons have typically been restricted to deep learning models, leaving traditional statistical and machine learning approaches unexamined even though they have compatible detection performance with deep learning models. To bridge this gap, we adopt FLOPs, a hardware-agnostic metric, and apply it to both traditional and deep learning models. This unified treatment enables fair and reproducible comparisons of computational cost across paradigms and provides practitioners with a principled basis for model selection in resource-constrained environments.

# B    ALGORITHMS AND DATASETS

## B.1    ALGORITHMS

**Hotelling**    (H.Hotelling, 1947). A multivariate statistical process control method that scores each observation via its Mahalanobis distance under a Gaussian reference model, thereby capturing correlated variation across variables.

**PCA**    (Shyu et al., 2003). Principal Component Analysis models normal structure in a low-dimensional subspace. Deviations are quantified through reconstruction error. Anomalies arise when observations project poorly onto the principal subspace.

**ABOD**    (Kriegel et al., 2008). Angle-Based Outlier Detection ranks points by the variance of angles formed with all other points, exploiting the geometric insight that outliers yield concentrated angle distributions in high dimensions. Practical variants use subsampling or $k$-nearest neighborhoods to reduce the quadratic cost while preserving discrimination.

**LOF**    (Breunig et al., 2000). Local Outlier Factor contrasts a point's local reachability density with that of its neighbors to assess how isolated it is within its immediate neighborhood. Large LOF

scores indicate locally sparse regions, enabling detection of context-dependent anomalies that global density models often miss.

**CBLOF** (He et al., 2003). Cluster-Based LOF assigns each point to a large or small cluster and computes scores from cluster size and distance to representative large clusters. Points in small, distant clusters receive high scores, capturing both rarity and separation.

**HBOS** (Goldstein & Dengel, 2012). Histogram-Based Outlier Score approximates feature-wise densities with univariate histograms and aggregates inverse densities across dimensions, implicitly assuming weak dependence.

**LODA** (Pevnỳ, 2016). The Lightweight online Detector of Anomalies ensembles sparse random projections, building one-dimensional histograms in each projected space and combining their anomaly evidences.

**Isolation Forest** (Liu et al., 2008). Random partitioning via isolation trees isolates anomalies with fewer splits, producing shorter expected path lengths than normal points. Scores are obtained by normalizing path lengths against the average in random trees, enabling fast, distribution-agnostic detection.

**HS-Tree** (Tan et al., 2011). Half-Space Trees construct randomized, axis-aligned partitions geared for streaming one-class detection. Points that consistently fall into underpopulated half-spaces obtain higher anomaly scores.

**DAGMM** (Zong et al., 2018). The Deep Autoencoding Gaussian Mixture Model jointly learns a compact representation and a GMM density in an end-to-end fashion, combining reconstruction features with mixture-based energy for scoring. This coupling allows the representation to align with density estimation, improving separability of rare patterns.

**DeepSVDD** (Ruff et al., 2018). A deep one-class objective trains a network to map normal data into a minimal radius hypersphere in latent space, penalizing distances from a fixed center. Samples that lie far from this center at test time are flagged as anomalies, avoiding reconstruction bias inherent to autoencoders.

**LSTM-AE** (Malhotra et al., 2016). A sequence-to-sequence LSTM autoencoder learns normal temporal dynamics and emits reconstruction errors over sliding windows. Sustained or abrupt increases in error indicate departures from learned patterns, capturing both gradual drifts and transient spikes.

**LSTM-VAE** (Park et al., 2018). A variational sequence model with LSTM encoder-decoder estimates a probabilistic generative process, enabling anomaly scoring via low evidence lower bound or high reconstruction error.

**USAD** (Audibert et al., 2020). A dual-autoencoder architecture trained with an adversarial-inspired objective where two decoders reconstruct each other's outputs to improve robustness. At inference, a calibrated combination of the two reconstruction errors yields stable anomaly scores with strong generalization across regimes.

**OmniAnomaly** (Su et al., 2019). A stochastic recurrent VAE augmented with normalizing flows models complex temporal dependencies and heteroscedastic noise, producing likelihood-based anomaly scores, negative log-probability, its latent dynamics capture both short and long range dependencies.

**LUAD** (Fan et al., 2023). A lightweight unsupervised detector that combines efficient temporal encoder, TCN, with a compact probabilistic module and an auxiliary diagnosis head.

Table 5: Dataset overview.

| Dataset | Dimensions | Train size | Test size | Anomaly Ratio (%) |
|---------|------------|------------|-----------|-------------------|
| PSM | 25 | 129784 | 87841 | 27.76 |
| MSL | 55 | 58317 | 73729 | 10.53 |
| SMAP | 25 | 135183 | 427617 | 12.79 |
| SMD | 38 | 708405 | 708420 | 4.16 |
| SWaT | 51 | 496800 | 449919 | 12.14 |
| WADI | 123 | 784537 | 172801 | 5.77 |

**Anomaly Transformer** Xu et al. (2022). Anomaly Transformer models time point relationships through a Gaussian prior and self attention based series association, using their association discrepancy as a discriminative anomaly score.

**TimesNet** Wu et al. (2023). TimesNet transforms one dimensional sequences into multi scale two dimensional periodic representations and learns temporal patterns through convolution blocks, providing string reconstruction features for anomaly detection.

### B.2 DATASETS

Table 5 summarizes the key statistics of the benchmark datasets, including the input dimension, the size of the train set, the size of the test set, and the anomaly ratio. Note that anomalies appear only in test sets, and the reported ratios are calculated with respect to the test instances. In addition to these summary statistics, we provide a description of the key characteristics of each dataset below.

**PSM** (Abdulaal et al., 2021). The Pooled Server Metrics dataset consists of multivariate time series monitoring server behavior, including signals such as CPU utilization and memory usage. It contains 13 weeks of training data and 8 weeks of testing data. Anomalies are present in both splits, while labels are provided only for the test set and include both planned and unplanned events.

**MSL** (Hundman et al., 2018). The Mars Science Laboratory dataset was constructed from telemetry of NASA's Curiosity rover. Anomalies were extracted from Incident Surprise, Anomaly reports (ISA) and manually labeled across channels.

**SMAP** (Hundman et al., 2018). The Soil Moisture Active Passive dataset was derived from telemetry collected during NASA's satellite mission. Anomalies were identified through ISA, which document unexpected spacecraft events during post-launch operations.

**SMD** (Su et al., 2019). The Server Machine Dataset is a 5 week multivariate time series collection gathered from a large Internet company. It comprises logs with metrics such as CPU load, memory usage, disk activity, and network traffic. The dataset is partitioned into training and testing halves, with anomalies in the testing portion labeled by domain experts based on incident reports.

**SWaT** (Mathur & Tippenhauer, 2016). The Secure Water Treatment dataset was collected from a fully operational 6 stage water treatment testbed. It comprises readings from sensors and actuators recorded every second over 11 consecutive days, including 7 days of normal operation and 4 days with controlled cyber-physical attacks. All attack instances were labeled by experts.

**WADI** (Ahmed et al., 2017). The Water Distribution dataset is derived from a scaled-down water distribution network testbed that simulates real industrial control systems. It contains multivariate time series of sensor and actuator signals across different stages of water storage and distribution. The dataset includes normal operations as well as periods with cyber-physical attack scenarios, with labels provided for the anomalous events.

## C   DERIVATIONS OF FLOPS

This section provides a detailed account of the FLOPs computations for both traditional and deep learning models. For traditional models, FLOPs are derived from the algorithmic procedures, with training and inference costs obtained by applying the formulas to the entire training and test sets, respectively. In contrast, deep learning models operate on sequences generated by a rolling window. To ensure consistency, all calculations used the same input shape, determined by the window length and feature dimension of each dataset. The FLOPs of deep learning models were computed using the `calflops` package, applied to our predefined model architectures. Training FLOPs were estimated as the sum of forward and backward passes, whereas inference FLOPs were measured from the forward pass alone. Thus, training FLOPs are calculated as the operations required to process one epoch of windowed training data with both forward and backward passes, while inference FLOPs correspond to a forward pass over the windowed test set. The full-training FLOPs are then obtained by multiplying the training FLOPs per epoch by the number of epochs used for each model. For comparability across models, all FLOPs are expressed in GFLOPs.

**Global notation and counting rule**   Given a common vector $x \in \mathbb{R}^d$, we use following notations: $n_{\text{tr}}$ denotes the number of training instances, $n_{\text{inf}}$ the number of inference instances, and $d$ the input dimensionality. Also, we denote FLOPs per sample by $f$, and the total FLOPs over the dataset by $F$. If an algorithm supports per-sample inference, we report both $f$ and $F$. For models that do not involve a distinct training phase, we set $n = n_{\text{tr}} = n_{\text{inf}}$.

Additions, multiplications, divisions, and comparisons are all counted as 1 FLOP, and the FLOPs required for matrix multiplication between $A \in \mathbb{C}^{M \times N}$ and $B \in \mathbb{C}^{N \times L}$ are calculated as

$$F_{AB} = 2MNL - ML$$

which consists of $MNL$ multiplication and $ML(N - 1)$ additions.

### C.1   HOTELLING

**Training FLOPs**   Hotelling's statistic is

$$T^2 = (x - \mu)^\top \Sigma^{-1} (x - \mu),$$

where $\mu \in \mathbb{R}^d$ and $\Sigma \in \mathbb{R}^{d \times d}$ are mean and covariance of the training set. In the training phase, the inverse covariance matrix is calculated.

(i) *Mean*: $\mu = \frac{1}{n_{\text{tr}}} \sum_i x_i$  costs

$$F_\mu = (n_{\text{tr}} - 1)d + d = n_{\text{tr}}d.$$

(ii) *Covariance*: $\Sigma = \frac{1}{n_{\text{tr}}-1} \sum_{i=1}^{n_{\text{tr}}} (x_i - \mu)(x_i - \mu)^\top$ costs

$$F_\Sigma = n_{\text{tr}}d + n_{\text{tr}}d^2 + (n_{\text{tr}} - 1)d^2 + d^2$$
$$= n_{\text{tr}}(2d^2 + d)$$

with respect to subtraction, multiplication, addition, and division.

(iii) *Inverse*: Inverting $\Sigma$ costs

$$F_{\Sigma^{-1}} \approx d^3.$$

Summing up,

$$F^{\text{train}} = F_\mu + F_\Sigma + F_{\Sigma^{-1}}$$
$$= 2n_{\text{tr}}d^2 + 2n_{\text{tr}}d + d^3.$$

**Inference FLOPs**   In the inference phase, the monitoring statistics $T^2$ calculations are performed sample by sample.

(i) *Centering subtraction*: $v = x - \mu$ costs

$$f_{\text{cen}} = d.$$

(ii) *Matrix multiplication*: $T^2 = v^\top \Sigma^{-1} v$ costs

$$f_{\text{mat}} = 2d^2 + d - 1.$$

Therefore,

$$f^{\text{infer}} = f_{\text{cen}} + f_{\text{mat}} = 2d^2 + 2d - 1,$$
$$F^{\text{infer}} = n_{\text{inf}}(2d^2 + 2d - 1).$$

## C.2 PCA

**Model-specific notation** $p$: number of principal components retained ($p \le d$).

**Training FLOPs** We estimate the mean and covariance exactly as in section C.1, and then obtain the top-$p$ eigenspace with QR decomposition.

(i) *Mean*: $\mu = \frac{1}{n_{\text{tr}}} \sum_i x_i$ costs

$$F_\mu = (n_{\text{tr}} - 1)d + d = n_{\text{tr}}d.$$

(ii) *Covariance*: $\Sigma = \frac{1}{n_{\text{tr}}-1} \sum_i^{n_{\text{tr}}} (x_i - \mu)(x_i - \mu)^\top$ costs

$$F_\Sigma \approx n_{\text{tr}}(2d^2 + d).$$

(iii) *QR decomposition of matrix*: Computing QR decomposition costs

$$F_{\text{dec}} \approx 3d^2.$$

Summing up,

$$F^{\text{train}} = F_\mu + F_\Sigma + F_{\text{dec}}$$
$$= 2n_{\text{tr}}d^2 + 2n_{\text{tr}}d + 3d^2.$$

**Inference FLOPs** In the inference phase, reconstructions can be performed sample by sample. Given $x \in \mathbb{R}^d$, let $U_p \in \mathbb{R}^{d \times p}$ be the loading matrix.

(i) *Projection*: $z = U_p^\top x$ costs

$$f_{\text{proj}} = p(2d - 1).$$

(ii) *Reconstruction*: $\hat{x} = U_p z$ costs

$$f_{\text{rec}} = d(2p - 1).$$

(iii) *Error calculation*: $e = \|x - \hat{x}\|^2 = \sum_{j=1}^d (x_j - \hat{x}_j)^2$ costs

$$f_{\text{err}} = 3d - 1.$$

Therefore, for each instance and total $n_{\text{inf}}$ instances,

$$f^{\text{infer}} = f_{\text{proj}} + f_{\text{err}} + f_{\text{rec}} = 4pd - p + 2d - 1,$$

$$F^{\text{infer}} = n_{\text{inf}}(4pd - p + 2d - 1).$$

## C.3 ABOD

**Model-specific notation** $k$: number of neighbors for Fast-ABOD.

**Training/Inference FLOPs** We first prepare $k$-NN neighborhoods, then evaluate the Angle Based Outlier Factor (ABOF) score for a point using its $k$ neighbors. This is Fast-ABOD which approximate original ABOD.

(i) *All-pairs of Euclidean distances*: Computing $\|x_i - x_j\|_2 = \sqrt{\sum_{l=1}^d (x_{il} - x_{jl})^2}$ costs $3d$. If span this to all-pairs, it costs

$$F_{\text{dist}} = \binom{n}{2}(3d) = \frac{3}{2}n(n-1)d.$$

(ii) *Sorting distances*: Sorting algorithms have approximated complexity of $O(n \log_2 n)$. Therefore, sorting all-pair distances costs

$$f_{\text{sort}} \approx (n-1) \log_2(n-1), \quad F_{\text{sort}} \approx n(n-1) \log_2(n-1).$$

(iii) *ABOF calculation*: ABOF is calculated by $\text{VAR}_{B,C \in N_k(A)}\left(\frac{\langle \overline{AB}, \overline{AC} \rangle}{\|\overline{AB}\|^2 \|\overline{AC}\|^2}\right)$ and $\binom{k}{2} = \frac{1}{2}k(k-1)$ is the number of neighbor pair cases.

Reusing calculated distances, each pair needs one dot product and operations for multiplication and normalization. Therefore, each pair needs $2d + 1$ FLOPs and spans $\frac{1}{2}k(k-1)$ times within one sample.

With $\frac{1}{2}k(k-1)$ pairs, $\text{VAR} = \frac{1}{N}\sum v^2 - \left(\frac{1}{N}\sum v\right)^2$ costs $\frac{3}{2}k(k-1) + 1$.

Then,

$$f_{\text{abof}} = \frac{1}{2}k(k-1)(2d+1) + \frac{3}{2}k(k-1) + 1 = k(k-1)(d+2) + 1$$

and

$$F_{\text{abof}} = nk(k-1)(d+2) + n.$$

Summing up,

$$\begin{aligned} F^{\text{train/infer}} &= F_{\text{dist}} + F_{\text{sort}} + F_{\text{abof}} \\ &= 1.5n(n-1)d + n(n-1)\log_2(n-1) + nk(k-1)(d+2) + n. \end{aligned}$$

## C.4 LOF

**Model-specific notation** $k$: number of neighbors.

**Training/Inference FLOPs** We build $k$-NN neighborhoods and then compute reachability distances, local reachability density, and the LOF score.

(i) *All-pairs of Euclidean distances*: As calculated at Section C.3, it costs

$$F_{\text{dist}} = \binom{n}{2}(3d) = \frac{3}{2}n(n-1)d.$$

(ii) *Sorting distances*: Sorting algorithms have approximated complexity of $O(n \log_2 n)$. Therefore, sorting all-pair distances costs

$$f_{\text{sort}} \approx (n-1) \log_2(n-1), \quad F_{\text{sort}} \approx n(n-1) \log_2(n-1).$$

(iii) *Reachability distances*: For each point $x_p$ and $x_o \in N_k(x_p)$, comparison operation $\text{reach\_dist}(x_p, x_o) = \max\{\text{dist}_k(x_o), \text{dist}(x_p, x_o)\}$ is conducted. Total comparison costs

$$f_{\text{reach}} = k, \quad F_{\text{reach}} = nk.$$

(iv) *Local reachability density (LRD)*: Formulation of LRD is

$$\text{LRD}(x_p) = \left(\frac{1}{N_k(x_p)} \sum_{x_o \in N_k(x_p)} \text{reach\_dist}(x_p, x_o)\right)^{-1}.$$

Per point, $k - 1$ additions, 1 division, and 1 scaling is conducted with total $k + 1$ FLOPs.

Therefore,

$$f_{\text{lrd}} = k + 1, \quad F_{\text{lrd}} = n(k+1).$$

(v) *LOF score*: Formulation of LOF score is

$$\text{LOF}(x_p) = \frac{1}{k} \sum_{x_o \in N_k(x_p)} \frac{\text{LRD}(x_o)}{\text{LRD}(x_p)}.$$

Per point, $k$ divisions, $k - 1$ additions, 1 division is conducted with total $2k$ FLOPs.

Therefore,

$$f_{\text{lof}} = 2k, \quad F_{\text{lof}} = 2nk.$$

Summing up,

$$\begin{aligned} F^{\text{train/infer}} &= F_{\text{dist}} + F_{\text{sort}} + F_{\text{reach}} + F_{\text{lrd}} + F_{\text{lof}} \\ &= 1.5n(n-1)d + n(n-1)\log_2(n-1) + nk + n(k+1) + 2nk. \end{aligned}$$

## C.5 CBLOF

**Model-specific notation**   $I$: maximum k-means iterations, $C$: number of clusters, $L$: number of large clusters, $|LC|$: number instances in large clusters.

**Training/Inference FLOPs**   CBLOF fits $C$ centroids with k-means and then scores samples using the large-small cluster partition.

(i) *K-means cluster assignment*: For each sample $x \in \mathbb{R}^d$ and each centroid $c$, squared distance $\|x-c\|^2$ costs $3d-1$ FLOPs. Also, finding minimum distance centroid over $C$ centroids contributes $C-1$ comparisons.

Over $n$ points and one iteration, it costs

$$F_{\text{assign/iter}} = n\left[C(3d-1) + (C-1)\right].$$

(ii) *K-means centroid update*: For each centroid, we accumulate assigned points and normalize once. As we have $n$ points and $C$ centroids, accumulation costs $d(n-C)$ and normalization costs $Cd$ making total FLOPs for centroid update is $nd$.

Over $n$ points and one iteration, it costs

$$F_{\text{update/iter}} = nd.$$

Therefore,

$$F_{\text{kmeans}} = I(F_{\text{assign/iter}} + F_{\text{update/iter}})$$
$$= I(3Cdn - n + nd).$$

(iii) *Scoring with large/small partition*: For each point $x_p$, score is computed as

$$Score(x_p) = \begin{cases} |C_i| \times \min_{j \in LC} \text{dist}(x_p, C_j), & \text{if } x_p \in C_i, C_i \in SC \text{ and } C_j \in LC \\ |C_i| \times \text{dist}(x_p, C_i), & \text{if } x_p \in C_i, \text{ and } C_i \in LC \end{cases}$$

where $C_i$ denotes $i^{th}$ cluster and $|C_i|$ denotes the number of points in each cluster. Also, $|LC|$ is the number of instances that belong to large clusters and $|SC|$ is the number of instances that belong to small clusters, formally, $|SC| = n - |LC|$.

If $p \in SC$: As $L$ large clusters exist, calculating distances to $L$ cluster centroid costs $L(3d-1)$ and comparing costs $L-1$. Therefore, each point in small cluster need $3dL$ FLOPs, including multiplication operation of cluster size.

If $p \in LC$: Calculating distances to their own centroid costs $3d-1$ and multiplication costs $1$ FLOPs. Therefore, each point in large cluster need $3d$ FLOPs.

Total scoring FLOPs for all $n$ samples is

$$F_{\text{score}} = 3d(|SC| \cdot L + |LC|)$$
$$= 3d\big((n - |LC|)\, L + |LC|\big).$$

Combining k-means and scoring,

$$F^{\text{train/infer}} = nI(3Cd + d - 1) + 3d((n - |LC|)L + |LC|).$$

## C.6 HBOS

**Model-specific notation**   $b$: number of bins per feature.

**Training FLOPs**   The training cost consists of histogram construction. Each sample-feature value is assigned to a bin with one subtraction and one division, giving $2n_{\text{tr}}d$ FLOPs in total. Converting counts to densities, computing bin widths, and performing the normalization check together require $5bd$ FLOPs.

Thus,

$$F^{\text{train}} = 2n_{\text{tr}}d + 5bd.$$

**Inference FLOPs**   At inference time, the score of each sample is computed based on the histograms. Each feature requires the computation of $\log_2(\text{hist} + \alpha)$, which costs 2 FLOPs per bin and yields $2bd$ FLOPs in total. For every sample and feature, assigning the score with boundary checks adds about 3 FLOPs, giving $3n_{\text{inf}}d$ FLOPs.

Therefore,

$$F^{\text{infer}} = 3n_{\text{inf}}d + 2bd.$$

## C.7   LODA

**Model-specific notation**   $b$: number of bins, $c$: number of random projections.

**Training FLOPs**   The training cost consists of sparse random projections and histogram construction, covering both the computation of projected values and the assignment of sample to bins for density estimation.

(i) *Sparse projection*: Each projection vector has $\sqrt{d}$ nonzero entries. Computing one projection value $z_{ij} = x_j^\top w_i$ requires $2\sqrt{d} - 1$ FLOPs. With $n_{\text{tr}}$ training samples and $c$ projections, the cost is

$$F_{\text{proj}} = n_{\text{tr}}c(2\sqrt{d} - 1).$$

(ii) *Histogram construction*: Each projected value must be assigned to a histogram bin. Using binary search over the $b$ bin edges requires $\log_2 b$ comparisons per assignment. The cost is therefore

$$F_{\text{bin}} = n_{\text{tr}}c \log_2 b.$$

Summing up, the training FLOPs are

$$F^{\text{train}} = F_{\text{proj}} + F_{\text{bin}}$$
$$= n_{\text{tr}}c(2\sqrt{d} + \log_2 b - 1).$$

**Inference FLOPs**   During inference, each sample is projected onto the $c$, its bin is determined, and the corresponding density values are used to compute the anomaly score.

(i) *Sparse projection*: Each projection requires $2\sqrt{d} - 1$ FLOPs, and the total projection cost over all histogram is

$$f_{\text{proj}} = c(2\sqrt{d} - 1).$$

(ii) *Bin lookup and score computation*: For each projection, locating the appropriate bin via binary search and computing log-density with accumulation require $c \log_2 b + 2$ FLOPs. Over all projections this becomes

$$f_{\text{binscore}} = c(\log_2 b + 2).$$

Therefore, including the final division for averaging across projections, the inference cost is

$$f^{\text{infer}} = f_{\text{proj}} + f_{\text{binscore}}$$
$$= 2c\sqrt{d} + c \log_2 b + c + 1,$$

$$F^{\text{infer}} = n_{\text{inf}}(2c\sqrt{d} + c \log_2 b + c + 1).$$

## C.8   ISOLATION FOREST

**Model-specific notation**   $T$: number of trees, $s$: max samples per tree, $\gamma$: Euler-Mascheroni constant ($\gamma \approx 0.5772$).

**Training FLOPs**   Each tree is grown on a random subsample of size $s$. At each internal node, we pick a random feature, sample a split value within the feature's range, and partition the instances by comparison. Let $n_l$ denote the expected number of samples at a node in level $l$. At a level $l$ node, computing feature's range costs $n_l$, and partitioning the instances costs $n_l$, since the tree is binary.

Also, we approximate the tree height as $h \approx \log_2 s$, the number of nodes in level $l$ as $2^l$, and number of samples processed in a level $l$ node as $n_l \approx \frac{s}{2^l}$. Therefore, computation at each node in level $l$ costs $2 \times \frac{s}{2^l}$. The total cost for a single tree is

$$\sum_{l=0}^{h-1} \left(2 \times \frac{s}{2^l} \times 2^l\right) = 2sh = 2s \log_2 s.$$

For $T$ trees the training FLOPs are approximated as

$$F^{\text{train}} = T(2s \log_2 s).$$

**Inference FLOPs**   A sample is routed from the root to a leaf in every tree. The expected path length $c(s)$ for subsample size $s$ is presented by authors of Isolation Forest (Liu et al., 2008).

$$c(s) = 2H_{s-1} - \frac{2(s-1)}{s} \approx 2(\ln(s-1) + \gamma) - 2 + \frac{2}{s}$$

Each step down the tree costs one comparison, hence costs per sample across $T$ trees are $T \cdot c(s)$.

Therefore,

$$f_{\text{step\_down}} = T \cdot c(s) = T\left[2\{\ln(s-1) + \gamma\} - 2(1 - \frac{1}{s})\right]$$

Anomaly score of the model is calculated by $2^{-\frac{\mathbb{E}(h(x))}{c(s)}}$.

Score calculation is performed by aggregating path lengths across the $T$ trees. This requires $T-1$ additions, followed by a normalization step that introduces two more scalar operations, division and exponentiation. We thus fold these into an overall $T + 2$ overhead.

Therefore, per sample cost is

$$f^{\text{infer}} = T \cdot c(s) + (T+2) = T\left[2\ln(s-1) + 2\gamma - 2 + \frac{2}{s}\right] + (T+2).$$

For $n_{\text{inf}}$ instances,

$$F^{\text{infer}} = n_{\text{inf}}(T \cdot c(s) + (T+2))$$
$$= n_{\text{inf}}\left(T\left[2\ln(s-1) + 2\gamma - 2 + \frac{2}{s}\right] + (T+2)\right).$$

### C.9   HS-TREE

**Model-specific notation**   $T$: number of trees,  $h$: maximum depth of tree,  $\psi$: reference window size.

**Training FLOPs**   Let $|Node| = \sum_{l=0}^{h} 2^l = 2^{h+1} - 1$ be the number of nodes in a full binary tree of height $h$. Each tree is built by updating simple per-node statistics and routing the $\psi$ reference samples level by level.

(i) *Per-node statistic updates*: For every node we find work range, yielding a constant cost of about 5 FLOPs per node.

Therefore, with a single tree,

$$F_{\text{stat/tree}} = 5(2^{h+1} - 1).$$

(ii) *Routing the $\psi$ reference samples*: At level $l$ there are $2^l$ nodes and, on average, each processes $\psi/2^l$ samples. With one comparison per routed sample, the cost per level is $\psi$.

Across levels,

$$F_{\text{routing/tree}} = \sum_{l=0}^{h} \psi \ = \ \psi(h+1).$$

Total FLOPs calculated is

$$F^{\text{train}} = T(F_{\text{stat/tree}} + F_{\text{routing/tree}})$$
$$= T(5(2^{h+1} - 1) + \psi(h+1)).$$

**Inference FLOPs** For each sample we route to level $h$ and calculate score, and update leaf mass statistics.

(i) *Path routing and scoring*: Per sample, one comparison per level is conducted with $h$ levels and scoring is conducted with the cumulative sum of $\text{Node.r} \times 2^{\text{Node.k}}$ which costs 3 FLOPs.

Thus,

$$f_{\text{routing/tree}} = h + 3.$$

(ii) *On-path mass updates*: Along the visited path, we update mass with 4 operations including two comparison for lower and upper bound, addition of count, and depth comparison.

As each point pass $h + 1$ nodes,

$$f_{\text{update/tree}} = 4(h + 1).$$

Therefore,

$$f^{\text{infer}} = T(f_{\text{routing/tree}} + f_{\text{update/tree}})$$
$$= T(5h + 7),$$

$$F^{\text{infer}} = n_{\text{inf}}T(5h + 7).$$

## D  DETAILED EXPERIMENT SETTINGS

**Hyperparameter Tuning** For both traditional and deep learning models, we defined hyperparameter search spaces based on ranges commonly adopted in prior benchmark studies. For each model, the parameters of each algorithm (e.g. number of estimators, number of bins, neighborhood size, clustering parameters, latent dimensions, dropout rates, training epochs) were specified as candidate sets. These search spaces were predetermined and systematically explored through grid search across all datasets. For each dataset and model pair, all hyperparameter combinations were evaluated, and the configuration yielding the highest AUROC was selected as the final setting. This procedure ensured that every model was tuned in a consistent and performance-oriented manner while remaining faithful to the parameter ranges established in the literature.

## E  ADDITIONAL EXPERIMENT RESULTS

### E.1  ADDITIONAL RESULTS OF SECTION 4.1

To complement the AUROC-based summary in Table 3, we further provide an additional table that reorganizes the results from the perspective of computational efficiency, which is presented in Table 6. Specifically, for each dataset, we first report the top five models ranked in ascending order of total FLOPs, allowing a direct comparison of which approaches remain feasible under strict resource budgets.

Following these efficiency-focused tables, we provide the complete results for each dataset. The following tables report AUROC, AUPRC, VUS-ROC, VUS-PR, and FLOPs across all evaluated models. We observe that GFLOPs for deep learning models are generally much larger than those of traditional models, despite yielding comparable accuracy. For clarity, the best performance score in each table is highlighted in **bold**, while the second-best score is underlined.

Table 6: Top five models for each dataset ranked in ascending order of total FLOPs. All computational costs are reported in gigaFLOPs (GFLOPs). AUROC and AUPRC scores are reported as the mean over five runs with different random seeds.

| Dataset | Model | Type | GFLOPs ↓ | | | AUROC ↑ | AUPRC ↑ |
|---|---|---|---|---|---|---|---|
| | | | Train | Inference | Full-training | | |
| PSM | HBOS | Statistical | 0.01 | 0.01 | — | $0.73 \pm 0.00$ | $0.50 \pm 0.00$ |
| | Isolation Forest | One-class | $< 0.01$ | 0.07 | — | $0.70 \pm 0.02$ | $0.47 \pm 0.03$ |
| | HS-Tree | One-class | $< 0.01$ | 0.10 | — | $0.54 \pm 0.02$ | $0.32 \pm 0.02$ |
| | LODA | Statistical | 0.13 | 0.10 | — | $0.65 \pm 0.03$ | $0.45 \pm 0.03$ |
| | Hotelling | Statistical | 0.17 | 0.11 | — | $0.77 \pm 0.00$ | $0.49 \pm 0.00$ |
| MSL | HBOS | Statistical | 0.01 | 0.01 | — | $0.62 \pm 0.00$ | $0.16 \pm 0.00$ |
| | HS-Tree | One-class | $< 0.01$ | 0.08 | — | $0.64 \pm 0.03$ | $0.14 \pm 0.03$ |
| | Isolation Forest | One-class | $< 0.01$ | 0.14 | — | $0.62 \pm 0.01$ | $0.15 \pm 0.01$ |
| | LODA | Statistical | 0.15 | 0.21 | — | $0.47 \pm 0.02$ | $0.11 \pm 0.02$ |
| | Hotelling | Statistical | 0.36 | 0.45 | — | $0.53 \pm 0.00$ | $0.13 \pm 0.00$ |
| SMAP | HBOS | Statistical | 0.01 | 0.03 | — | $0.61 \pm 0.00$ | $0.15 \pm 0.00$ |
| | LODA | Statistical | 0.08 | 0.31 | — | $0.45 \pm 0.09$ | $0.12 \pm 0.02$ |
| | Isolation Forest | One-class | $< 0.01$ | 0.48 | — | $0.64 \pm 0.01$ | $0.16 \pm 0.00$ |
| | PCA | Reconstruction | 0.18 | 0.36 | — | $0.41 \pm 0.00$ | $0.11 \pm 0.00$ |
| | HS-Tree | One-class | 0.01 | 0.70 | — | $0.45 \pm 0.01$ | $0.11 \pm 0.00$ |
| SMD | HBOS | Statistical | 0.05 | 0.08 | — | $0.63 \pm 0.00$ | $0.14 \pm 0.00$ |
| | HS-Tree | One-class | $< 0.01$ | 0.81 | — | $0.64 \pm 0.02$ | $0.07 \pm 0.01$ |
| | Isolation Forest | One-class | $< 0.01$ | 0.90 | — | $0.68 \pm 0.01$ | $0.16 \pm 0.01$ |
| | LODA | Statistical | 0.55 | 0.63 | — | $0.62 \pm 0.02$ | $0.11 \pm 0.01$ |
| | PCA | Reconstruction | 2.1 | 2.09 | — | $0.65 \pm 0.00$ | $0.11 \pm 0.00$ |
| SWaT | HBOS | Statistical | 0.05 | 0.07 | — | $0.85 \pm 0.00$ | $0.75 \pm 0.00$ |
| | Isolation Forest | One-class | $< 0.01$ | 0.36 | — | $0.83 \pm 0.00$ | $0.73 \pm 0.01$ |
| | HS-Tree | One-class | 0.01 | 0.74 | — | $0.37 \pm 0.08$ | $0.11 \pm 0.01$ |
| | LODA | Statistical | 1.39 | 1.39 | — | $0.82 \pm 0.02$ | $0.73 \pm 0.02$ |
| | Hotelling | Statistical | 2.64 | 2.39 | — | $0.65 \pm 0.00$ | $0.17 \pm 0.00$ |
| WADI | HS-Tree | One-class | $< 0.01$ | 0.20 | — | $0.63 \pm 0.05$ | $0.09 \pm 0.02$ |
| | HBOS | Statistical | 0.19 | 0.06 | — | $0.74 \pm 0.00$ | $0.18 \pm 0.00$ |
| | Isolation Forest | One-class | $< 0.01$ | 0.33 | — | $0.74 \pm 0.02$ | $0.18 \pm 0.01$ |
| | LODA | Statistical | 1.46 | 0.34 | — | $0.72 \pm 0.04$ | $0.27 \pm 0.05$ |
| | DeepSVDD | One-class | 19.5 | 1.43 | 4874.05 | $0.43 \pm 0.03$ | $0.05 \pm 0.00$ |

Table 7: All results on the PSM dataset with performance metrics and GFLOPs.

| Model | Type | GFLOPs ↓ | | | AUROC ↑ | AUPRC ↑ | VUS-ROC ↑ | VUS-PR ↑ |
|---|---|---|---|---|---|---|---|---|
| | | Train | Inference | Full-training | | | | |
| Hotelling | Statistical | 0.17 | 0.11 | — | $\mathbf{0.77 \pm 0.00}$ | $\underline{0.49 \pm 0.00}$ | $\mathbf{0.71 \pm 0.00}$ | $\mathbf{0.49 \pm 0.00}$ |
| ABOD | Statistical | 926.33 | 421.87 | — | $0.75 \pm 0.00$ | $0.43 \pm 0.00$ | $0.66 \pm 0.00$ | $0.45 \pm 0.00$ |
| LOF | One-class | 917.76 | 416.07 | — | $0.73 \pm 0.00$ | $0.42 \pm 0.00$ | $0.63 \pm 0.00$ | $0.43 \pm 0.00$ |
| CBLOF | One-class | 6.56 | 9.91 | — | $0.70 \pm 0.02$ | $0.45 \pm 0.01$ | $0.64 \pm 0.02$ | $0.44 \pm 0.01$ |
| PCA | Reconstruction | 0.17 | 0.16 | — | $0.65 \pm 0.00$ | $0.47 \pm 0.00$ | $0.59 \pm 0.00$ | $0.43 \pm 0.00$ |
| HBOS | Statistical | 0.01 | 0.01 | — | $0.73 \pm 0.00$ | $\mathbf{0.50 \pm 0.00}$ | $0.68 \pm 0.00$ | $0.48 \pm 0.00$ |
| LODA | Statistical | 0.13 | 0.10 | — | $0.65 \pm 0.03$ | $0.45 \pm 0.03$ | $0.58 \pm 0.03$ | $0.41 \pm 0.02$ |
| Isolation Forest | One-class | $< 0.01$ | 0.07 | — | $0.70 \pm 0.02$ | $0.47 \pm 0.03$ | $0.67 \pm 0.02$ | $0.46 \pm 0.02$ |
| HS-Tree | One-class | $< 0.01$ | 0.10 | — | $0.54 \pm 0.02$ | $0.32 \pm 0.02$ | $0.55 \pm 0.02$ | $0.34 \pm 0.02$ |
| DAGMM | Statistical | 5.31 | 1.20 | 530.62 | $0.50 \pm 0.03$ | $0.28 \pm 0.01$ | $0.47 \pm 0.03$ | $0.30 \pm 0.01$ |
| DeepSVDD | One-class | 1.57 | 0.35 | 156.20 | $0.69 \pm 0.01$ | $0.37 \pm 0.01$ | $0.57 \pm 0.01$ | $0.40 \pm 0.01$ |
| LSTM-AE | Reconstruction | 12.23 | 2.76 | 1223.50 | $\underline{0.76 \pm 0.01}$ | $0.50 \pm 0.04$ | $\underline{0.70 \pm 0.01}$ | $\underline{0.48 \pm 0.01}$ |
| LSTM-VAE | Reconstruction | 547.41 | 123.50 | 136852.56 | $0.61 \pm 0.06$ | $0.42 \pm 0.07$ | $0.57 \pm 0.06$ | $0.40 \pm 0.06$ |
| USAD | Reconstruction | 2.63 | 0.59 | 657.91 | $0.52 \pm 0.01$ | $0.34 \pm 0.01$ | $0.44 \pm 0.01$ | $0.32 \pm 0.00$ |
| OmniAnomaly | Reconstruction | 10.15 | 2.29 | 203.05 | $0.61 \pm 0.00$ | $0.44 \pm 0.00$ | $0.58 \pm 0.01$ | $0.41 \pm 0.00$ |
| LUAD | Reconstruction | 4956.20 | 1118.16 | 148685.87 | $0.67 \pm 0.01$ | $0.48 \pm 0.01$ | $0.62 \pm 0.01$ | $0.44 \pm 0.01$ |
| Anomaly Transformer | Reconstruction | 4965.57 | 1120.27 | 148967.04 | $0.52 \pm 0.01$ | $0.32 \pm 0.01$ | $0.56 \pm 0.00$ | $0.33 \pm 0.01$ |
| TimesNet | Reconstruction | 1380.51 | 311.45 | 27610.22 | $0.60 \pm 0.00$ | $0.40 \pm 0.00$ | $0.65 \pm 0.00$ | $0.45 \pm 0.00$ |

Table 8: All results on the MSL dataset with performance metrics and GFLOPs.

| Model | Type | GFLOPs ↓ | | | AUROC ↑ | AUPRC ↑ | VUS-ROC ↑ | VUS-PR ↑ |
|---|---|---|---|---|---|---|---|---|
| | | Train | Inference | Full-training | | | | |
| Hotelling | Statistical | 0.36 | 0.45 | – | 0.53 ± 0.00 | 0.13 ± 0.00 | 0.58 ± 0.00 | 0.15 ± 0.00 |
| ABOD | Statistical | 334.71 | 536.74 | – | 0.63 ± 0.00 | 0.17 ± 0.00 | 0.63 ± 0.00 | 0.18 ± 0.00 |
| LOF | One-class | 334.41 | 536.36 | – | 0.56 ± 0.00 | 0.12 ± 0.00 | 0.59 ± 0.00 | 0.14 ± 0.00 |
| CBLOF | One-class | 0.89 | 2.60 | – | **0.65 ± 0.01** | **0.20 ± 0.01** | 0.67 ± 0.01 | **0.20 ± 0.01** |
| PCA | Reconstruction | 0.36 | 0.64 | – | 0.53 ± 0.00 | 0.14 ± 0.00 | 0.59 ± 0.00 | 0.15 ± 0.00 |
| HBOS | Statistical | 0.01 | 0.01 | – | 0.62 ± 0.00 | 0.16 ± 0.00 | 0.66 ± 0.00 | 0.18 ± 0.00 |
| LODA | Statistical | 0.15 | 0.21 | – | 0.47 ± 0.02 | 0.11 ± 0.02 | 0.52 ± 0.03 | 0.12 ± 0.01 |
| Isolation Forest | One-class | < 0.01 | 0.14 | – | 0.62 ± 0.01 | 0.15 ± 0.01 | 0.66 ± 0.01 | 0.16 ± 0.01 |
| HS-Tree | One-class | < 0.01 | 0.08 | – | 0.64 ± 0.03 | 0.14 ± 0.03 | 0.68 ± 0.02 | 0.17 ± 0.02 |
| DAGMM | Statistical | 2.55 | 1.08 | 127.74 | 0.50 ± 0.02 | 0.13 ± 0.01 | 0.52 ± 0.01 | 0.12 ± 0.01 |
| DeepSVDD | One-class | 2.54 | 1.07 | 253.56 | 0.48 ± 0.03 | 0.13 ± 0.00 | 0.48 ± 0.02 | 0.12 ± 0.01 |
| LSTM-AE | Reconstruction | 22.77 | 9.60 | 455.49 | 0.56 ± 0.00 | 0.15 ± 0.00 | 0.61 ± 0.00 | 0.17 ± 0.00 |
| LSTM-VAE | Reconstruction | 21.23 | 8.95 | 5307.31 | 0.53 ± 0.00 | 0.13 ± 0.01 | 0.59 ± 0.00 | 0.15 ± 0.00 |
| USAD | Reconstruction | 2.07 | 0.87 | 206.77 | 0.44 ± 0.00 | 0.13 ± 0.00 | 0.47 ± 0.00 | 0.12 ± 0.00 |
| OmniAnomaly | Reconstruction | 6.25 | 2.63 | 124.94 | 0.53 ± 0.00 | 0.14 ± 0.00 | 0.59 ± 0.00 | 0.14 ± 0.00 |
| LUAD | Reconstruction | 2362.72 | 995.71 | 23627.17 | 0.50 ± 0.00 | 0.13 ± 0.00 | 0.56 ± 0.00 | 0.14 ± 0.00 |
| Anomaly Transformer | Reconstruction | 576.01 | 242.75 | 17280.39 | 0.51 ± 0.00 | 0.12 ± 0.00 | 0.54 ± 0.01 | 0.13 ± 0.00 |
| TimesNet | Reconstruction | 21851.23 | 9208.69 | 437024.60 | 0.64 ± 0.03 | 0.16 ± 0.01 | **0.68 ± 0.03** | 0.19 ± 0.01 |

Table 9: All results on the SMAP dataset with performance metrics and GFLOPs.

| Model | Type | GFLOPs ↓ | | | AUROC ↑ | AUPRC ↑ | VUS-ROC ↑ | VUS-PR ↑ |
|---|---|---|---|---|---|---|---|---|
| | | Train | Inference | Full-training | | | | |
| Hotelling | Statistical | 0.18 | 0.56 | – | 0.42 ± 0.00 | 0.11 ± 0.00 | 0.42 ± 0.00 | 0.11 ± 0.00 |
| ABOD | Statistical | 998.15 | 10281.98 | – | 0.64 ± 0.00 | 0.17 ± 0.00 | 0.58 ± 0.00 | 0.17 ± 0.00 |
| LOF | One-class | 996.78 | 10277.64 | – | 0.62 ± 0.00 | **0.17 ± 0.00** | 0.56 ± 0.00 | **0.17 ± 0.00** |
| CBLOF | One-class | 2.48 | 10.76 | – | 0.62 ± 0.01 | 0.16 ± 0.00 | 0.58 ± 0.01 | 0.16 ± 0.00 |
| PCA | Reconstruction | 0.18 | 0.36 | – | 0.41 ± 0.00 | 0.11 ± 0.00 | 0.42 ± 0.00 | 0.11 ± 0.00 |
| HBOS | Statistical | 0.01 | 0.03 | – | 0.61 ± 0.00 | 0.15 ± 0.00 | 0.57 ± 0.00 | 0.16 ± 0.00 |
| LODA | Statistical | 0.08 | 0.31 | – | 0.45 ± 0.09 | 0.12 ± 0.02 | 0.45 ± 0.08 | 0.12 ± 0.02 |
| Isolation Forest | One-class | < 0.01 | 0.48 | – | **0.64 ± 0.01** | 0.16 ± 0.00 | **0.59 ± 0.01** | 0.17 ± 0.00 |
| HS-Tree | One-class | 0.01 | 0.70 | – | 0.45 ± 0.01 | 0.11 ± 0.00 | 0.43 ± 0.01 | 0.12 ± 0.00 |
| DAGMM | Statistical | 5.53 | 5.83 | 552.69 | 0.51 ± 0.01 | 0.13 ± 0.00 | 0.51 ± 0.00 | 0.14 ± 0.00 |
| DeepSVDD | One-class | 2.73 | 2.88 | 272.94 | 0.60 ± 0.00 | 0.15 ± 0.00 | 0.55 ± 0.00 | 0.17 ± 0.00 |
| LSTM-AE | Reconstruction | 12.74 | 13.44 | 637.20 | 0.49 ± 0.03 | 0.12 ± 0.01 | 0.50 ± 0.03 | 0.13 ± 0.01 |
| LSTM-VAE | Reconstruction | 39.87 | 42.04 | 9966.77 | 0.44 ± 0.02 | 0.11 ± 0.00 | 0.45 ± 0.02 | 0.11 ± 0.00 |
| USAD | Reconstruction | 5.68 | 5.99 | 1419.32 | 0.41 ± 0.01 | 0.10 ± 0.00 | 0.41 ± 0.01 | 0.11 ± 0.00 |
| OmniAnomaly | Reconstruction | 34.43 | 36.30 | 688.56 | 0.47 ± 0.00 | 0.12 ± 0.00 | 0.49 ± 0.00 | 0.13 ± 0.00 |
| LUAD | Reconstruction | 5162.37 | 5443.28 | 51623.73 | 0.50 ± 0.01 | 0.12 ± 0.00 | 0.51 ± 0.01 | 0.13 ± 0.00 |
| Anomaly Transformer | Reconstruction | 254.83 | 268.70 | 7644.90 | 0.52 ± 0.02 | 0.13 ± 0.01 | 0.54 ± 0.02 | 0.14 ± 0.01 |
| TimesNet | Reconstruction | 1428.44 | 1506.16 | 28568.71 | 0.47 ± 0.00 | 0.11 ± 0.00 | 0.48 ± 0.00 | 0.12 ± 0.00 |

Table 10: All results on the SMD dataset with performance metrics and GFLOPs.

| Model | Type | GFLOPs ↓ | | | AUROC ↑ | AUPRC ↑ | VUS-ROC ↑ | VUS-PR ↑ |
|---|---|---|---|---|---|---|---|---|
| | | Train | Inference | Full-training | | | | |
| Hotelling | Statistical | 2.10 | 2.10 | – | 0.73 ± 0.00 | 0.16 ± 0.00 | 0.73 ± 0.00 | 0.14 ± 0.00 |
| ABOD | Statistical | 38426.94 | 38428.58 | – | 0.71 ± 0.00 | 0.10 ± 0.00 | 0.69 ± 0.00 | 0.10 ± 0.00 |
| LOF | One-class | 38357.60 | 38359.24 | – | 0.65 ± 0.00 | 0.07 ± 0.00 | 0.65 ± 0.00 | 0.08 ± 0.00 |
| CBLOF | One-class | 36.30 | 34.65 | – | 0.72 ± 0.01 | 0.16 ± 0.01 | 0.68 ± 0.01 | 0.12 ± 0.01 |
| PCA | Reconstruction | 2.10 | 2.09 | – | 0.65 ± 0.00 | 0.11 ± 0.00 | 0.67 ± 0.00 | 0.10 ± 0.00 |
| HBOS | Statistical | 0.05 | 0.08 | – | 0.63 ± 0.00 | 0.14 ± 0.00 | 0.62 ± 0.00 | 0.10 ± 0.00 |
| LODA | Statistical | 0.55 | 0.63 | – | 0.62 ± 0.02 | 0.11 ± 0.01 | 0.62 ± 0.02 | 0.09 ± 0.01 |
| Isolation Forest | One-class | < 0.01 | 0.90 | – | 0.68 ± 0.01 | 0.16 ± 0.01 | 0.66 ± 0.01 | 0.11 ± 0.00 |
| HS-Tree | One-class | < 0.01 | 0.81 | – | 0.64 ± 0.02 | 0.07 ± 0.01 | 0.64 ± 0.02 | 0.08 ± 0.01 |
| DAGMM | Statistical | 21.79 | 7.26 | 2178.62 | 0.53 ± 0.05 | 0.04 ± 0.01 | 0.53 ± 0.02 | 0.06 ± 0.01 |
| DeepSVDD | One-class | 5.34 | 1.78 | 1336.00 | 0.61 ± 0.02 | 0.11 ± 0.02 | 0.54 ± 0.01 | 0.08 ± 0.00 |
| LSTM-AE | Reconstruction | 82.70 | 27.57 | 8269.64 | 0.77 ± 0.01 | **0.18 ± 0.01** | 0.75 ± 0.00 | 0.15 ± 0.00 |
| LSTM-VAE | Reconstruction | 791.06 | 263.69 | 197764.01 | 0.63 ± 0.01 | 0.10 ± 0.00 | 0.62 ± 0.01 | 0.09 ± 0.00 |
| USAD | Reconstruction | 37.57 | 12.52 | 9392.39 | 0.39 ± 0.00 | 0.03 ± 0.00 | 0.41 ± 0.00 | 0.05 ± 0.00 |
| OmniAnomaly | Reconstruction | 198.12 | 66.04 | 3962.46 | 0.69 ± 0.02 | 0.11 ± 0.02 | 0.66 ± 0.02 | 0.11 ± 0.01 |
| LUAD | Reconstruction | 27766.94 | 9255.84 | 833008.11 | 0.67 ± 0.00 | 0.11 ± 0.00 | 0.67 ± 0.01 | 0.10 ± 0.00 |
| Anomaly Transformer | Reconstruction | 3545.62 | 1181.90 | 106368.46 | 0.52 ± 0.01 | 0.07 ± 0.00 | 0.63 ± 0.01 | 0.08 ± 0.00 |
| TimesNet | Reconstruction | 25325.13 | 8441.89 | 506502.64 | **0.77 ± 0.00** | 0.17 ± 0.00 | **0.79 ± 0.00** | **0.15 ± 0.00** |

Table 11: All results on the SWaT dataset with performance metrics GFLOPs.

| Model | Type | GFLOPs ↓ Train | GFLOPs ↓ Inference | GFLOPs ↓ Full-training | AUROC ↑ | AUPRC ↑ | VUS-ROC ↑ | VUS-PR ↑ |
|---|---|---|---|---|---|---|---|---|
| Hotelling | Statistical | 2.64 | 2.39 | − | $0.65 \pm 0.00$ | $0.17 \pm 0.00$ | $0.59 \pm 0.00$ | $0.17 \pm 0.00$ |
| ABOD | Statistical | 23615.66 | 19345.49 | − | $0.60 \pm 0.00$ | $0.15 \pm 0.00$ | $0.55 \pm 0.00$ | $0.15 \pm 0.00$ |
| LOF | One-class | 23551.21 | 19287.13 | − | $0.71 \pm 0.00$ | $0.32 \pm 0.00$ | $0.51 \pm 0.00$ | $0.21 \pm 0.00$ |
| CBLOF | One-class | 3.95 | 4.01 | − | $0.81 \pm 0.01$ | $0.70 \pm 0.01$ | $0.56 \pm 0.01$ | $0.34 \pm 0.00$ |
| PCA | Reconstruction | 2.64 | 2.42 | − | $0.82 \pm 0.00$ | $0.73 \pm 0.00$ | $0.61 \pm 0.00$ | $0.43 \pm 0.00$ |
| HBOS | Statistical | 0.05 | 0.07 | − | $\mathbf{0.85 \pm 0.00}$ | $\mathbf{0.75 \pm 0.00}$ | $0.68 \pm 0.00$ | $\underline{0.48 \pm 0.00}$ |
| LODA | Statistical | 1.39 | 1.39 | − | $0.82 \pm 0.02$ | $0.73 \pm 0.02$ | $0.66 \pm 0.04$ | $0.47 \pm 0.04$ |
| Isolation Forest | One-class | < 0.01 | 0.36 | − | $\underline{0.83 \pm 0.00}$ | $\underline{0.73 \pm 0.01}$ | $\underline{0.70 \pm 0.01}$ | $0.48 \pm 0.01$ |
| HS-Tree | One-class | 0.01 | 0.74 | − | $0.37 \pm 0.08$ | $0.11 \pm 0.01$ | $0.37 \pm 0.07$ | $0.11 \pm 0.01$ |
| DAGMM | Statistical | 20.24 | 6.11 | 2023.86 | $0.51 \pm 0.00$ | $0.12 \pm 0.00$ | $0.51 \pm 0.01$ | $0.13 \pm 0.00$ |
| DeepSVDD | One-class | 5.11 | 1.54 | 1277.45 | $0.82 \pm 0.03$ | $0.72 \pm 0.03$ | $0.59 \pm 0.10$ | $0.37 \pm 0.07$ |
| LSTM-AE | Reconstruction | 187.15 | 56.50 | 9357.33 | $0.76 \pm 0.01$ | $0.28 \pm 0.03$ | $0.61 \pm 0.01$ | $0.25 \pm 0.01$ |
| LSTM-VAE | Reconstruction | 584.52 | 176.45 | 146130.74 | $0.76 \pm 0.11$ | $0.59 \pm 0.26$ | $0.58 \pm 0.08$ | $0.37 \pm 0.13$ |
| USAD | Reconstruction | 32.79 | 9.90 | 3278.73 | $0.81 \pm 0.00$ | $0.69 \pm 0.01$ | $0.54 \pm 0.00$ | $0.33 \pm 0.01$ |
| OmniAnomaly | Reconstruction | 51.30 | 15.49 | 1026.05 | $0.83 \pm 0.00$ | $0.73 \pm 0.00$ | $\mathbf{0.78 \pm 0.01}$ | $\mathbf{0.56 \pm 0.01}$ |
| LUAD | Reconstruction | 5653.12 | 1706.55 | 169593.69 | $0.75 \pm 0.01$ | $0.25 \pm 0.01$ | $0.66 \pm 0.03$ | $0.26 \pm 0.02$ |
| Anomaly Transformer | Reconstruction | 19166.41 | 5785.92 | 574992.39 | $0.44 \pm 0.07$ | $0.21 \pm 0.04$ | $0.43 \pm 0.07$ | $0.15 \pm 0.01$ |
| TimesNet | Reconstruction | 280014.48 | 84530.22 | 5600289.64 | $0.28 \pm 0.00$ | $0.09 \pm 0.00$ | $0.30 \pm 0.00$ | $0.09 \pm 0.00$ |

Table 12: All results on the WADI dataset with performance metrics GFLOPs.

| Model | Type | GFLOPs ↓ Train | GFLOPs ↓ Inference | GFLOPs ↓ Full-training | AUROC ↑ | AUPRC ↑ | VUS-ROC ↑ | VUS-PR ↑ |
|---|---|---|---|---|---|---|---|---|
| Hotelling | Statistical | 23.93 | 5.27 | − | $0.53 \pm 0.00$ | $0.06 \pm 0.00$ | $0.53 \pm 0.00$ | $0.06 \pm 0.00$ |
| ABOD | Statistical | 125696.96 | 6047.49 | − | $0.49 \pm 0.00$ | $0.12 \pm 0.00$ | $0.44 \pm 0.00$ | $0.07 \pm 0.00$ |
| LOF | One-class | 125611.74 | 6028.72 | − | $0.54 \pm 0.00$ | $0.09 \pm 0.00$ | $0.48 \pm 0.00$ | $0.08 \pm 0.00$ |
| CBLOF | One-class | 97.02 | 1.66 | − | $0.53 \pm 0.01$ | $0.21 \pm 0.00$ | $0.46 \pm 0.01$ | $0.11 \pm 0.00$ |
| PCA | Reconstruction | 23.93 | 3.18 | − | $0.50 \pm 0.00$ | $0.05 \pm 0.00$ | $0.46 \pm 0.00$ | $0.06 \pm 0.00$ |
| HBOS | Statistical | 0.19 | 0.06 | − | $\mathbf{0.74 \pm 0.00}$ | $0.18 \pm 0.00$ | $0.67 \pm 0.00$ | $0.17 \pm 0.00$ |
| LODA | Statistical | 1.46 | 0.34 | − | $0.72 \pm 0.04$ | $\mathbf{0.27 \pm 0.05}$ | $\mathbf{0.68 \pm 0.04}$ | $\mathbf{0.22 \pm 0.04}$ |
| Isolation Forest | One-class | < 0.01 | 0.33 | − | $\underline{0.74 \pm 0.02}$ | $0.18 \pm 0.01$ | $0.67 \pm 0.01$ | $0.16 \pm 0.01$ |
| HS-Tree | One-class | < 0.01 | 0.20 | − | $0.63 \pm 0.05$ | $0.09 \pm 0.02$ | $0.62 \pm 0.06$ | $0.10 \pm 0.02$ |
| DAGMM | Statistical | 150.17 | 11.03 | 15017.03 | $0.45 \pm 0.05$ | $0.05 \pm 0.01$ | $0.40 \pm 0.06$ | $0.05 \pm 0.01$ |
| DeepSVDD | One-class | 19.50 | 1.43 | 4874.05 | $0.43 \pm 0.03$ | $0.05 \pm 0.00$ | $0.37 \pm 0.04$ | $0.05 \pm 0.00$ |
| LSTM-AE | Reconstruction | 490.76 | 36.03 | 49075.61 | $0.54 \pm 0.00$ | $0.21 \pm 0.01$ | $0.49 \pm 0.00$ | $0.11 \pm 0.01$ |
| LSTM-VAE | Reconstruction | 1031.52 | 75.73 | 257880.45 | $0.54 \pm 0.01$ | $0.19 \pm 0.03$ | $0.49 \pm 0.01$ | $0.12 \pm 0.01$ |
| USAD | Reconstruction | 146.25 | 10.74 | 36562.76 | $0.45 \pm 0.01$ | $0.20 \pm 0.00$ | $0.40 \pm 0.01$ | $0.10 \pm 0.00$ |
| OmniAnomaly | Reconstruction | 358.51 | 26.32 | 7170.23 | $0.58 \pm 0.00$ | $\underline{0.24 \pm 0.00}$ | $0.51 \pm 0.00$ | $0.14 \pm 0.00$ |
| LUAD | Reconstruction | 36349.70 | 2668.78 | 1090490.86 | $0.54 \pm 0.00$ | $0.21 \pm 0.00$ | $0.49 \pm 0.00$ | $0.11 \pm 0.00$ |
| Anomaly Transformer | Reconstruction | 8076.86 | 593.00 | 242305.76 | $0.55 \pm 0.03$ | $0.09 \pm 0.01$ | $0.56 \pm 0.02$ | $0.07 \pm 0.00$ |
| TimesNet | Reconstruction | 24899.00 | 1828.07 | 497979.94 | $0.66 \pm 0.01$ | $0.20 \pm 0.00$ | $\underline{0.67 \pm 0.01}$ | $\underline{0.19 \pm 0.00}$ |

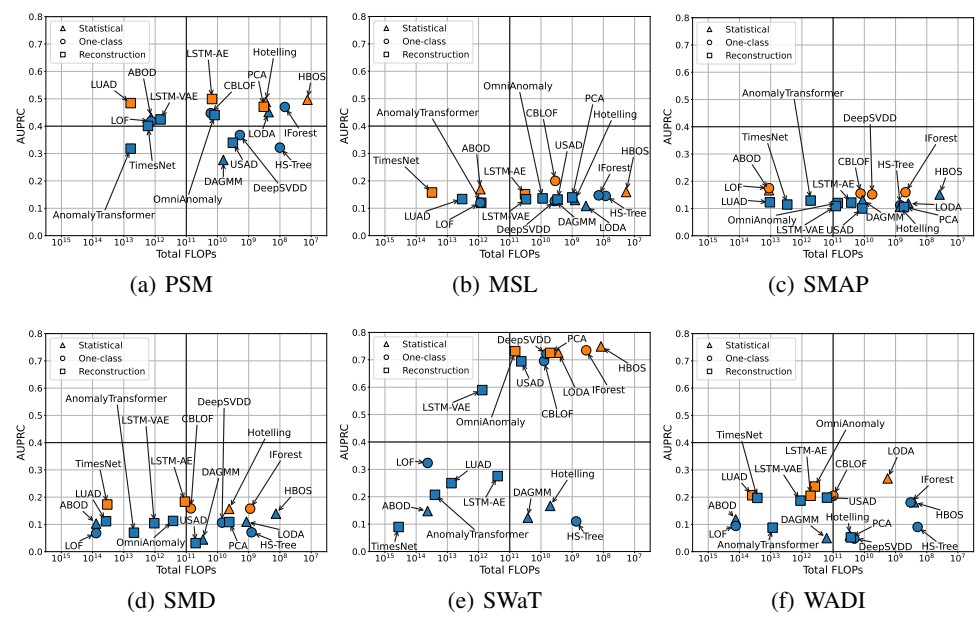

(a) PSM  (b) MSL  (c) SMAP

(d) SMD  (e) SWaT  (f) WADI

Figure 4: AUPRC vs. Total FLOPs (sum of training and inference FLOPs) for each dataset. Orange markers denote the top five models in terms of AUPRC.

## E.2 ADDITIONAL RESULTS OF SECTION 4.2

For each dataset, we report the estimated execution time under Highly Resourced, Mobile, and Edge scenarios. The following figures show dataset specific visualizations that highlight differences in model feasibility across environments. These results confirm the overall trend that deep learning models demand substantial resources, while traditional models remain efficient, although the degree of variation differs across datasets.

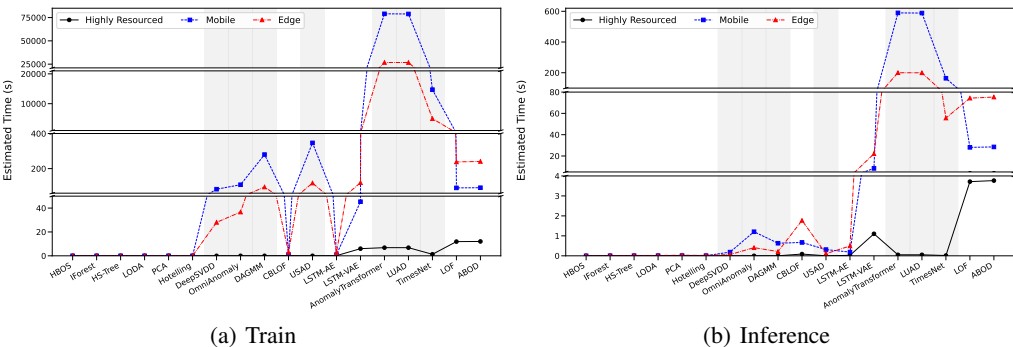

(a) Train             (b) Inference

Figure 5: Estimated execution time on PSM.

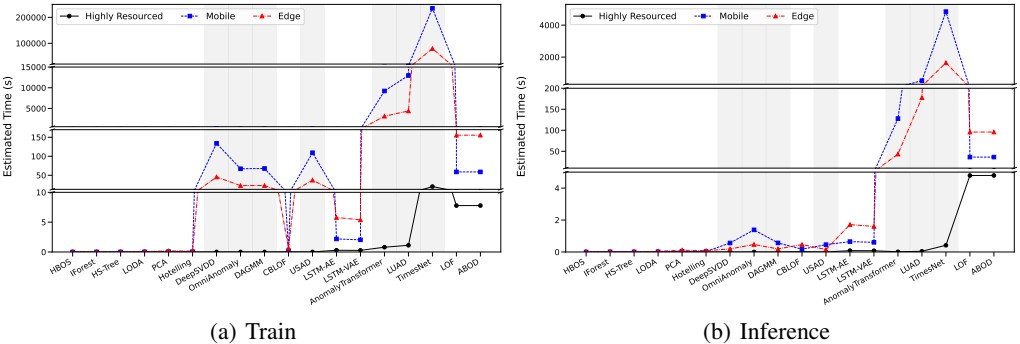

(a) Train             (b) Inference

Figure 6: Estimated execution time on MSL.

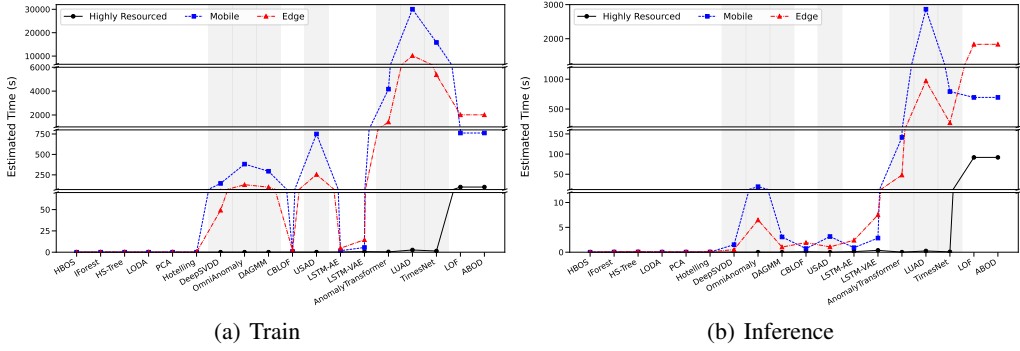

(a) Train             (b) Inference

Figure 7: Estimated execution time on SMAP.

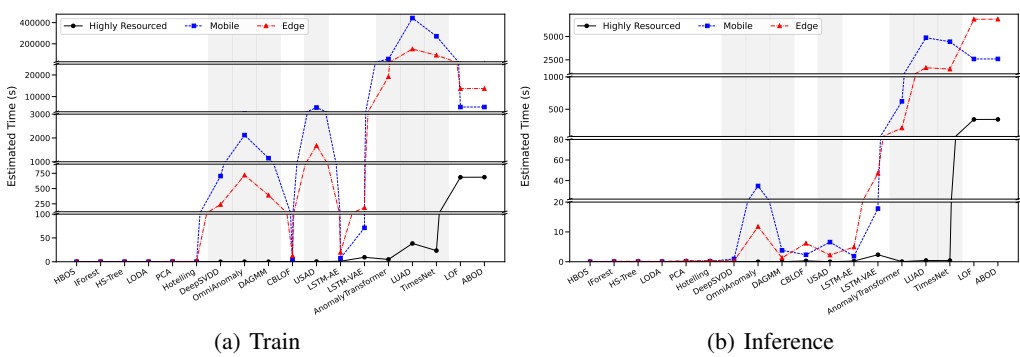

(a) Train

(b) Inference

Figure 8: Estimated execution time on SMD.

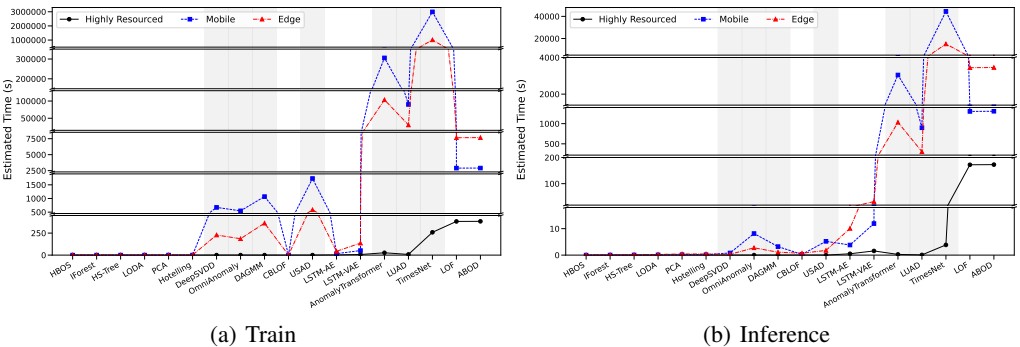

(a) Train

(b) Inference

Figure 9: Estimated execution time on SWaT.

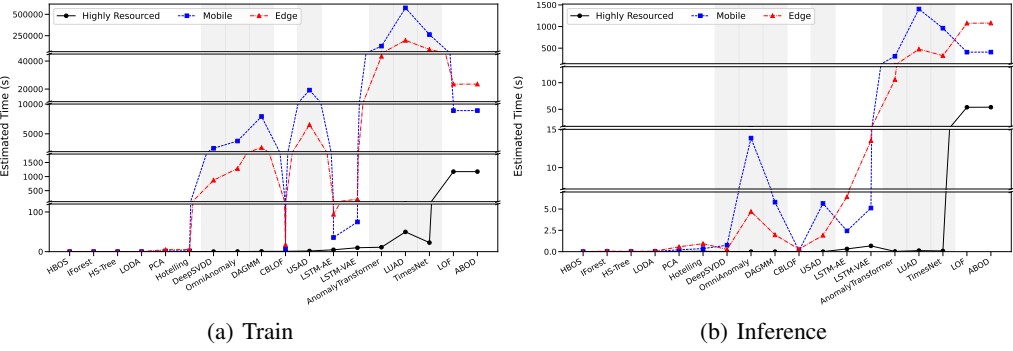

(a) Train

(b) Inference

Figure 10: Estimated execution time on WADI.

### E.3 ADDITIONAL RESULTS OF SECTION 4.3

We report scalability tests conducted across datasets of varying data size and dimensionality. For the size scalability experiment, FLOPs were measured while progressively slicing the length of the dataset, with the maximum test range determined by the smaller of the training and inference sets to ensure comparability between the two phases. For the dimension scalability experiment, FLOPs were measured while progressively increasing the feature dimension by sampling features in ten percent increments of the full dimensionality for each dataset. All models evaluated in this study are included. The left panel of each figure depicts training FLOPs, while the right panel presents inference FLOPs, with both y-axes plotted on a logarithmic scale. Overall, the results reveal consistent scaling patterns across datasets. $k$-NN based methods exhibit steep growth in computational cost as data size or dimension increases, while tree-based and projection methods remain relatively efficient.

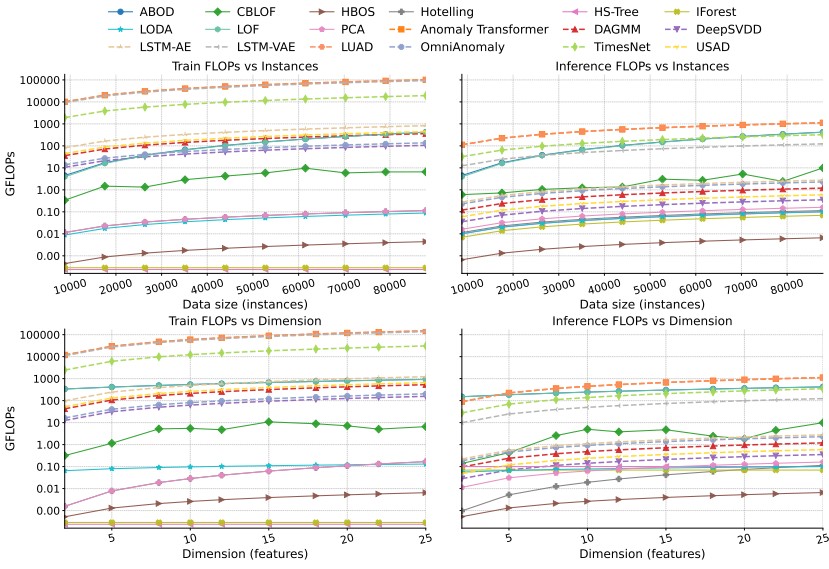

Figure 11: Scalability results on PSM.

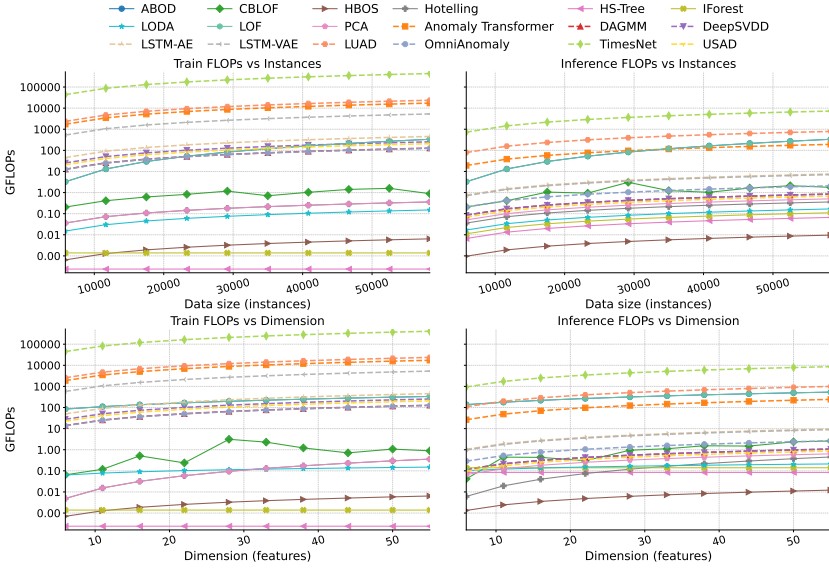

Figure 12: Scalability results on MSL.

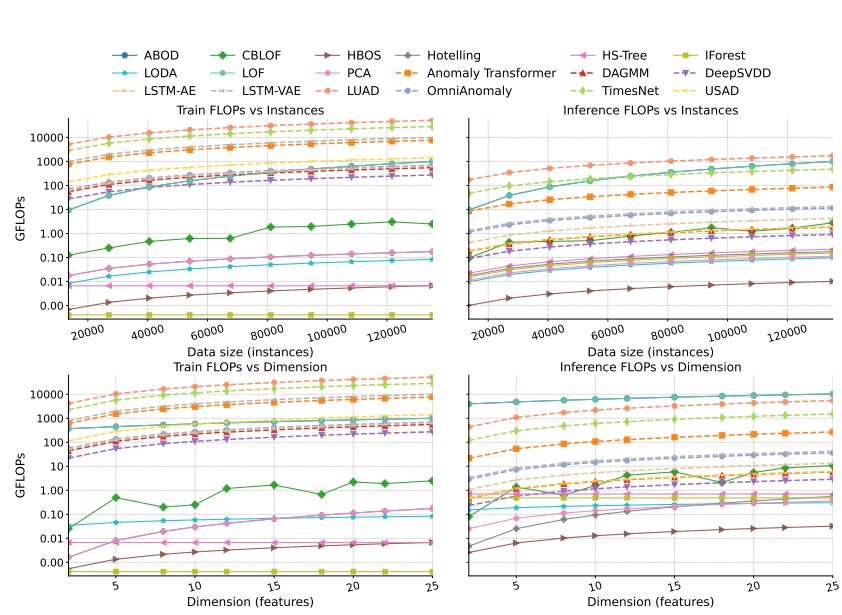

Figure 13: Scalability results on SMAP.

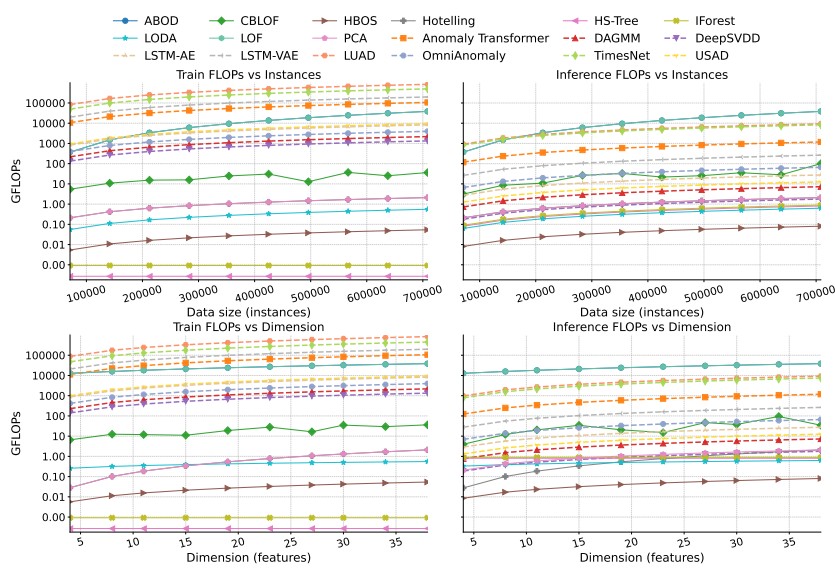

Figure 14: Scalability results on SMD.

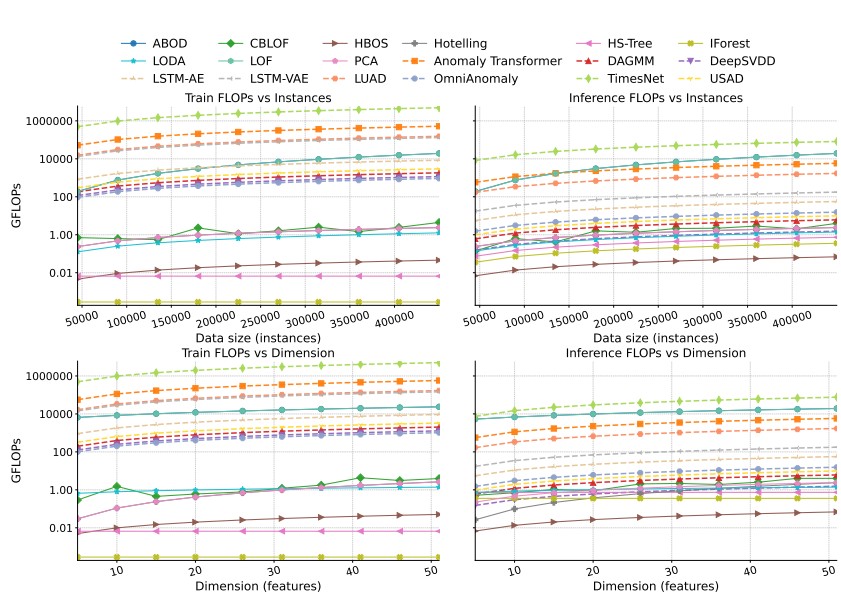

Figure 15: Scalability results on SWaT.

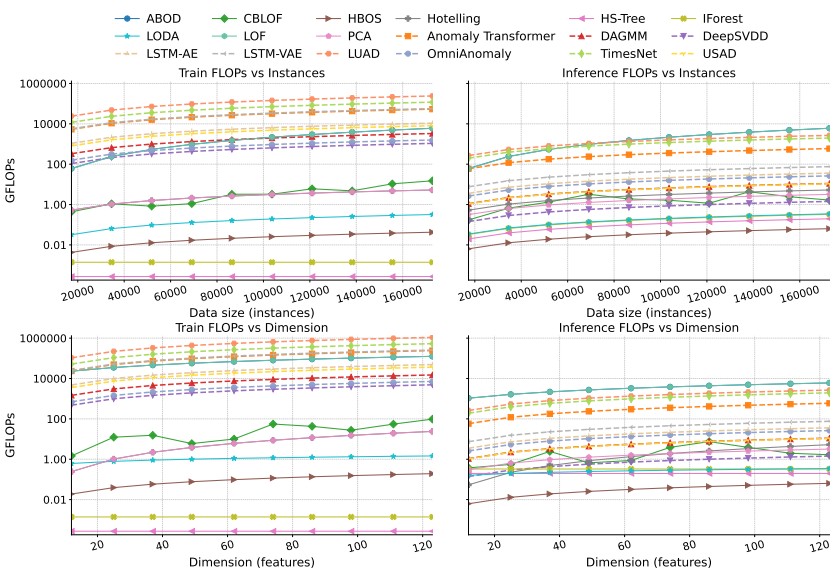

Figure 16: Scalability results on WADI.

### E.4 LLM USAGE STATEMENT

The large language model (LLM) was used solely to improve the clarity and readability of the manuscript. Specifically, they helped polish the writing, refine grammar, and improve phrasing. The use of LLM was limited to language editing, and LLM did not contribute to the ideation of the research, experimental design, implementation, analysis, or interpretation of the results.

