# OpenReview forum: "Rethinking Heavy Models in Multivariate Time Series Anomaly Detection"
_ICLR.cc/2026/Conference — Submitted to ICLR 2026_

### Official Review · Reviewer_HTcA · 2025-10-29

**Soundness:** 3
**Presentation:** 3
**Contribution:** 2
**Rating:** 4
**Confidence:** 3

**Summary:**

The paper studies multi-variate time series anomaly detection in the context of accuracy and compute efficiency.  It carries out a comparison of statistical, classical machine learning, and deep learning based methods through the lens of efficiency and finds that often non-deep learning alternatives perform better under limited compute budgets.

**Strengths:**

It is a well-written paper that poses an important question and seeks its answer by putting together a systematic study. Section 3.2.1 outlines the principles and reasoning that guided the selection of anomaly detection methods and benchmarks used in this study.  The paper uses FLOPs as a measure of computational resources required by a method and AUROC as the measure of performance.  Table 3 that captures performance vs. accuracy supports the assertion that deep learning model, despite having large computational requirements, rarely achieve performance that is "far better" than what is achieved by other techniques.

**Weaknesses:**

I don't quite get Figure 2.  Especially, why the individual dots connected by dashed lines ... aren't these discrete measurements?

AUROC is useful but insufficient for deployment.  We require thre threshold protocol, calibration and robustness to threshold selection, and domain specific accuracy requirements in order to decide which of the given list of methods is most useful in a given setting.  I would hazard a guess while the results are insightfull, these lack actionable information that a practitioned may be able to use to deploy a multi-variate time-series anomaly detection in industrial settings.

**Questions:**

Most anomaly detection methods require some sort of a threshold to make the final estimation whether or not an anomaly as occured.  This work seems to downplay the effect of threshold selection on the overall accuracy.  How AUROC is computed in the absence of threshold, or how thresholds were selected (where needed)?  Is it possible that some methods are less sensitive to threshold selection?  Perhaps this is a factor that should also be taken into account in a study such as the one proposed in this paper?

Imagine we are given a method that has very low compute requirements as measured by FLOPs, but also very low accuracy as measured by AUROC.  Where would this method sit within the comparisons presented in Table 3?  Also, when we search for acceptable performance vs. accuracy balance, we often have a some minimum accuracy requirements, which may be different in different domains.  How would that play out in the recommendations outlined in this work?

---

> ### Author Response · Authors · 2025-11-21
> **Response to Reviewer HTcA — Weakness #1**
>
> Thank you for pointing out important questions.
>
> ## Weakness \#1
> > I don't quite get Figure 2. Especially, why the individual dots connected by dashed lines ... aren't these discrete measurements?
>
> For Figure 2, the plotted values are indeed discrete measurements for each model. The reason the points are visually connected with dashed lines is simply because we adopted a line-plot style to improve readability under extreme scale differences across models and hardware settings. The dotted segments do not imply continuity or interpolation, and they have no analytical meaning beyond helping the reader visually trace the trend.
> The primary challenge was that training and inference times vary by several orders of magnitude across the evaluated models. To present all results within a single unified figure, while still making the magnitude differences interpretable, we used vertically stacked sub-axes, each having a different y-axis scale. This layout, combined with connecting the discrete points, allows readers to quickly see
> (1) how large the disparities are
> (2) where each model falls relative to others.
> If we used only isolated points without connecting lines, many models with small values would become visually indistinguishable at this resolution.
>
> For easier understanding, as an example, TimesNet requires only about 60 seconds to train under the Highly Resourced scenario, but the same model requires nearly 600,000 seconds under the Mobile scenario. These values are discrete measurements, yet the difference is so large that plotting them as isolated points would make the smaller values nearly invisible.

---

> ### Author Response · Authors · 2025-11-21
> **Response to Reviewer HTcA — Weakness #2 / Question #1**
>
> ## Weakness \#2 \& Question \#1
> Since both weakness and question are pointing about the AUROC metric and threshold selection, we will discuss them together in this part.
>
> You mentioned at weakness \#2 and question \#1,
> > AUROC is useful but insufficient for deployment. We require threshold protocol, calibration and robustness to threshold selection, and domain specific accuracy requirements in order to decide which of the given list of methods is most useful in a given setting. This work seems to downplay the effect of threshold selection on the overall accuracy.
>
> We thank you for highlighting the importance of threshold selection and its role in real-world deployments. We fully agree that practical anomaly detection systems ultimately rely on a threshold, and that threshold choice affects the final detection outcome.
>
> Our study therefore adopts AUROC as the primary evaluation metric because it provides a threshold-free and model-agnostic comparison. AUROC summarizes performance across all possible thresholds, avoiding unfairness introduced by model-specific calibration rules and enabling consistent comparison across heterogeneous model families (statistical, one-class, and deep learning models). AUROC also naturally reflects robustness to threshold choice. Models that are highly sensitive to threshold selection tend to yield flatter, lower ROC curves.
>
> Regarding your question on threshold selection, all methods output continuous anomaly scores, and AUROC is computed directly from these scores without applying any thresholding or heuristic calibration. This allows us to compare intrinsic ranking quality independent of domain-specific criteria. Practitioners can later select thresholds appropriate for their operating regime (considering TPR-FPR trade-off) directly from the ROC curve.

---

> ### Author Response · Authors · 2025-11-21
> **Response to Reviewer HTcA — Question #2**
>
> ## Question \#2
> > Imagine we are given a method that has very low compute requirements as measured by FLOPs, but also very low accuracy as measured by AUROC. Where would this method sit within the comparisons presented in Table 3? Also, when we search for acceptable performance vs. accuracy balance, we often have a some minimum accuracy requirements, which may be different in different domains. How would that play out in the recommendations outlined in this work?
>
> In Table 3, we report only the top-five models per dataset based on AUROC, chosen from the full set of evaluated methods. Consequently, a method that exhibits very low AUROC, regardless of its computational efficiency, would simply not appear in Table 3, because it would not fall within the AUROC-based top-five.
>
> This design is intentional. In many practical applications, practitioners must satisfy domain-specific minimum accuracy requirements before considering computational efficiency. Table 3 is therefore structured to be most useful in such scenarios. By listing only the highest-performing models for each dataset, the table enables practitioners to (1) identify models that achieve acceptable accuracy and (2) then choose the models with the lowest FLOPs among those. This helps reveal models that are not only accurate but also computationally attractive, and highlights candidates that are “universally strong” across datasets.
>
> We also acknowledge there must be the opposite case that domains with hard resource constraints, as you mentioned, practitioners may need to filter models by FLOPs first. To support this use case, we have added an additional table in the Appendix listing the top-five models per dataset ranked by lower FLOPs, enabling readers to first screen methods by computational budget and then examine their corresponding AUROC.
>
> We include the complete set of AUROC (along with additional performance metrics, AUPRC, VUS-ROC, VUS-PR in response to the requests from reviewer YBTB and reviewer dAaG) and FLOPs results for all models is included in the Appendix Section E.1. We appreciate your thoughtful question and have clarified these points in the revised manuscript.

---

### Official Review · Reviewer_PNpt · 2025-10-31

**Soundness:** 3
**Presentation:** 3
**Contribution:** 3
**Rating:** 4
**Confidence:** 4

**Summary:**

This work focuses on the problem of multivariate anomaly detection in limited-resource scenarios, as encountered in many real-world applications. To that end, this experimental work proposes to answer the following two questions:

- What are the most effective options for time series anomaly detection under limited computational resources, and are deep learning methods always the best options?
- Does a trade-off between detection performance and computational cost truly exist in practice?

To answer the questions, the paper carries three different experiments to assess performance vs efficiency: 1) trade-off between detection accuracy, measured by AUROC, and the computational cost quantified by training and inference FLOPs; 2) FLOPs (algorithmic operation counts) vs FLOPS (hardware throughput comparison; and 3) scalability as a function of data volume.

**Strengths:**

- This is a well-written paper with a clear experimental setup that aims to address a practical, but relevant question.
- Good coverage of baselines
- Insightful conclusions

**Weaknesses:**

Overall, this is a good paper. I see as a weakness that this may not be the typical paper expected in ICLR (which motivates my score), but I do not see major weaknesses for an experimental paper.

- As hardware is central to this work, it would have been good to that the different hardware configuration is reported in the main paper.

**Questions:**

Similar efforts [1], though not focused on resource constraints, have investigated the advantages of traditional vs. deep learning based approaches. Could you position your work with respect to this one and establish similarities and differences?

[1] https://doi.org/10.1016/j.patcog.2022.108945

---

> ### Author Response · Authors · 2025-11-21
> **Response to Reviewer PNpt — Weakness #1**
>
> Thank you for your positive assessment and for highlighting these concerns. We respectfully would like to clarify why we believe our paper fits well within the scope of ICLR.
>
> ## Weakness \#1
> > I see as a weakness that this may not be the typical paper expected in ICLR.
>
> First, although ICLR is often associated with theoretical or algorithmic innovation, the conference has consistently published experiment-driven papers that benchmark diverse models under unified evaluation protocols. Examples include:
> [1] Goswami, Mononito, et al. "Unsupervised Model Selection for Time Series Anomaly Detection." The Eleventh International Conference on Learning Representations. 2023.
> [2] Han, Xiaotian, et al. "FFB: A Fair Fairness Benchmark for In-Processing Group Fairness Methods." The Twelfth International Conference on Learning Representations. 2024.
> [3] Liu, Yantao, et al. "RM-Bench: Benchmarking Reward Models of Language Models with Subtlety and Style." The Thirteenth International Conference on Learning Representations. 2025.
> [4] Chan, Jun Shern, et al. "MLE-bench: Evaluating Machine Learning Agents on Machine Learning Engineering." The Thirteenth International Conference on Learning Representations. 2025.
>
> These works show that comprehensive empirical benchmarking is well aligned with ICLR’s standards.
>
> Second, ICLR has also accepted many papers where hardware considerations, computational efficiency, or resource constraints are central motivations. For example,
> [5] Xu, Zhijian, Ailing Zeng, and Qiang Xu. "FITS: Modeling Time Series with $10 k $ Parameters." The Twelfth International Conference on Learning Representations. 2024.
> [6] Fox, Sean, et al. "A block minifloat representation for training deep neural networks." International Conference on Learning Representations. 2020.
> [7] Tay, Yi, et al. "Long Range Arena: A Benchmark for Efficient Transformers." International Conference on Learning Representations. 2021.
>
> Our work follows this line of research: it challenges the prevailing assumption that deep learning models are always superior for time-series anomaly detection and highlights the importance of computation-cost-aware model selection in resource-constrained environments.
>
> > It would have been good to that the different hardware configuration is reported in the main paper.
>
> Finally, regarding hardware details, our paper already evaluates models under three different hardware scenarios and reports their estimated runtimes in Section 4.2. However, we agree that explicitly stating hardware configurations improves clarity, and we have added detailed hardware specifications and real training/inference times for representative models in Section 4.2.

---

> ### Author Response · Authors · 2025-11-21
> **Response to Reviewer PNpt — Question #1**
>
> ## Question \#1
> > Similar efforts [1], though not focused on resource constraints, have investigated the advantages of traditional vs. deep learning based approaches. Could you position your work with respect to this one and establish similarities and differences?
> [1] https://doi.org/10.1016/j.patcog.2022.108945
>
> Our work shares similarities with [1] in that both studies evaluate a diverse set of traditional and deep learning methods on widely used multivariate time-series anomaly detection datasets. In that sense, both papers contribute to understanding the relative detection performance across methodological families.
> However, our work differs from [1] in several key aspects:
> - **Explicit focus on computational efficiency and resource constraints**
> While [1] primarily analyzes detection accuracy, our study is centered on the performance-computational cost trade-off, with fine-grained FLOPs estimates for training and inference. This allows us to concretely illustrate one of the major limitations of deep learning approaches, their substantially higher computational cost, and to characterize when such models may or may not be appropriate under realistic resource budgets. This dimension is not addressed in [1].
> - **Actionable guidance for practitioners under domain-specific constraints**
> Rather than only concluding that deep learning does not always outperform traditional methods (a finding also observed in [1]), our study systematically identifies which models are preferable under accuracy requirements, resource constraints, or domain-specific deployment regimes (e.g. highly resourced servers vs. mobile or edge devices). We therefore complement the insights of [1] with prescriptive recommendations grounded in computational feasibility.
> - **Hardware-Agnostic Complexity Analysis**
> In addition, our work provides an important contribution that prior studies do not offer. Existing benchmarks lack a truly hardware-agnostic computational cost comparison between traditional and deep models. To address this, we derive closed-form FLOPs formulas for all traditional methods, enabling a unified and fair computation-based comparison across model families. This analytic contribution makes it possible to assess performance and computational cost trade-offs on equal terms, which was not feasible with prior runtime-based evaluations.
>
> Together, these differences position our work as a computationally grounded extension of prior comparative studies such as [1]. By integrating both detection efficacy and computational cost, our study provides a more comprehensive and deployment-oriented understanding of traditional and deep learning approaches.

---

> > ### Comment · Reviewer_PNpt · 2025-11-27
> >
> > Thanks for your answers and for providing pointers to works that are similar in nature to yours. This clarifies my main concern.

---

> > > ### Author Response · Authors · 2025-11-28
> > >
> > > We are pleased that our response addressed your main concern. Thank you for taking the time to review our paper.

---

### Official Review · Reviewer_dAaG · 2025-11-01

**Soundness:** 2
**Presentation:** 2
**Contribution:** 2
**Rating:** 2
**Confidence:** 4

**Summary:**

This paper revisits the necessity of heavy deep learning architectures for multivariate time series anomaly detection under resource-constrained environments. By introducing a hardware-agnostic measure (FLOPs) as a proxy for computational cost, the study contributes to the discussion on the effectiveness-efficiency trade-off in anomaly detection.

**Strengths:**

* The use of a hardware-agnostic FLOPs measure for runtime evaluation is well-motivated and provides a fair framework for cross-model comparison.
* The derivation and formulation of FLOPs for popular TSAD algorithms are clearly described.
* The paper presents a comprehensive evaluation of various models with respect to FLOPs, offering insights into the trade-offs between performance and computational efficiency.

**Weaknesses:**

* The study is constrained by limited datasets, algorithms, and evaluation aspects, which may restrict the generalizability of its conclusions.
* Limited theoretical or diagnostic interpretation of accuracy-runtime trade-off between lightweight and heavy models.

Please find the detailed comments in the following section.

**Questions:**

* Limited TSAD algorithm coverage. The study includes only 16 anomaly detection algorithms, while recent benchmark efforts (e.g., Schmidl et al., 2022 with 70 methods; Liu & Paparrizos, 2024 with 40 methods) have evaluated much larger and more diverse collections.
* Limited evaluation measures. Relying solely on AUROC overlooks known limitations of this measure in time series anomaly detection (e.g., its sensitivity to temporal noise, imbalance in anomaly ratio, and inability to capture temporal localization). Recent works have proposed time-series–aware measures such as Range-F1 [1] and VUS-PR [2], which should be considered for a more robust evaluation.
* Dataset selection limitations. The benchmark omits widely used datasets such as Exathlon [3] and TSB-AD (Liu & Paparrizos, 2024), which represent more diverse operational scenarios and anomaly types.
* The study mainly investigates multivariate time series but does not address the univariate case. Furthermore, it is unclear how model scalability behaves with respect to both sequence length and feature dimensionality, as no explicit analysis of these factors is provided.
* While adopting FLOPs as a hardware-agnostic proxy is a good practice, complementing it with real runtime measurements and memory footprint analyses across identical hardware would provide stronger evidence for practical applicability in real deployments.
* The paper would benefit from a deeper analysis of when and why lightweight or heavy models perform better. For instance, cases where LSTM-VAE achieves relatively low FLOPs but competitive performance compared to heavier models like LOF need further investigation. Beyond aggregate AUROC scores across the entire dataset, it would be useful to analyze performance across different anomaly types and anomaly ratios to better contextualize the observed trends.

[1] Tatbul N, Lee TJ, Zdonik S, Alam M, Gottschlich J. Precision and recall for time series. Advances in neural information processing systems. 2018;31.

[2] Paparrizos J, Boniol P, Palpanas T, Tsay RS, Elmore A, Franklin MJ. Volume under the surface: a new accuracy evaluation measure for time-series anomaly detection. Proceedings of the VLDB Endowment. 2022 Jul 1;15(11):2774-87.

[3] Jacob V, Song F, Stiegler A, Rad B, Diao Y, Tatbul N. Exathlon: a benchmark for explainable anomaly detection over time series. Proceedings of the VLDB Endowment. 2021 Jul 1;14(11):2613-26.

---

> ### Author Response · Authors · 2025-11-21
> **Response to Reviewer dAaG — Weakness #1 / Question #1, #2**
>
> Thank you for giving us valuable questions.
>
> ## Weakness \#1
> > The study is constrained by limited datasets, algorithms, and evaluation aspects, which may restrict the generalizability of its conclusions.
>
> We agree that broader datasets, algorithms, and evaluation dimensions could further improve generalizability. However, the intention of the study is not to provide an exhaustive benchmark, but rather to investigate two focused research questions. The experimental design was therefore tailored to the scope of these questions, aiming to understand the trade-off between detection performance and computational cost under representative and resource-constrained settings.
>
> Our goal is to extract clear empirical insights rather than to enumerate all possible datasets or anomaly detection methods. While the study does not attempt to cover the entire landscape of time series anomaly detection, the selected datasets and algorithms span diverse domains and modeling paradigms, which we believe are sufficient to support the conclusions regarding the stated research questions.
>
> Nonetheless, we carefully reviewed the comment and updated the manuscript to incorporate the suggested improvements and provide additional context where appropriate. We have also provided detailed responses to each of your questions below.
>
> ## Question \#1
> > Limited TSAD algorithm coverage.
>
> As noted in our response to Weakness 1, the scope of this study is centered on answering the research questions rather than maximizing algorithmic coverage. To make the experimental design meaningful for our objective, the selection of TSAD models followed the three principles described in Section 3.2.1:
> 1. Chronological coverage: including representative methods from early multivariate statistical monitoring to contemporary deep learning approaches.
> 2. Diversity of mechanisms: covering statistical, one-class, and reconstruction-based families to capture fundamentally different operating principles.
> 3. Baseline significance: prioritizing models that have been well established and widely adopted in prior TSAD studies.
>
> Based on the criteria listed above, we believe that this selection is sufficiently representative for answering the research questions of this study. Nonetheless, for more generalizability, we incorporated recent Transformer-based approaches, including Anomaly Transformer and TimesNet, into the revised manuscript.
>
>
> ## Question \#2
> > Limited evaluation measures.
>
> Our choice of AUROC was intentional, as threshold-dependent metrics often exhibit significant variation depending on how thresholds are selected across models. To compare methods fairly and isolate intrinsic model performance, we therefore adopted AUROC as a threshold-free measure. We acknowledge, however, that AUROC alone may not fully capture temporal localization or the effects of imbalanced anomaly ratios in time series settings. In response to your comment, we incorporated recently proposed evaluation metrics, including VUS-AUC and VUS-PR, which explicitly account for collective anomalies and temporal ranges. Additionally, AUPRC has been added for completeness. These results are now reported in the Section 4 and Appendix Section E.1. They support the same conclusions as our AUROC-based analysis.

---

> ### Author Response · Authors · 2025-11-21
> **Response to Reviewer dAaG — Question #3**
>
> ## Question \#3
> > Dataset selection limitations.
>
> We acknowledge that additional datasets such as Exathlon and TSB-AD provide complementary scenarios. However, our focus is on understanding trade-off between effectiveness and efficiency under representative settings commonly adopted in prior TSAD work, rather than providing an exhaustive benchmark across all available datasets. To support this choice, we note that the datasets used in our study (MSL, SMD, SWaT, SMAP, WADI, and PSM) correspond closely to those employed in many state-of-the-art multivariate TSAD research works:
> - OmniAnomaly (2019) [1]: *MSL, SMD, SWaT, SMAP, WADI*
> - USAD (2020) [2]: *MSL, SMD, SMAP*
> - Anomaly Transformer (2022) [3]: *MSL, SMD, SWaT, SMAP, PSM*
> - TranAD (2022) [4]: *MSL, SMD, SWaT, SMAP, WADI,* MSDS, NAB, UCR, MBA
> - TimesNet (2023) [5]:  *MSL, SMD, SWaT, SMAP, PSM*
> - ImDiffusion (2023) [6]: *MSL, SMD, SWaT, SMAP, PSM,* GCP
> - LUAD (2023) [7]: *MSL, SMD, SMAP*
> - PUAD (2024) [8]: *MSL, SMD, SMAP, PSM,* DND
> - TopoGDN (2024) [9]: *MSL, SMD, SWaT, WADI*
>
> These works collectively demonstrate that the selected datasets form the commonly used datasets suite for multivariate TSAD, covering diverse operational domains such as spacecraft telemetry, industrial control systems, and sensor-based cyber-physical systems.
>
> [1] Su, Ya, et al. "Robust anomaly detection for multivariate time series through stochastic recurrent neural network." Proceedings of the 25th ACM SIGKDD international conference on knowledge discovery & data mining. 2019.
> [2] Audibert, Julien, et al. "Usad: Unsupervised anomaly detection on multivariate time series." Proceedings of the 26th ACM SIGKDD international conference on knowledge discovery & data mining. 2020.
> [3] Xu, Jiehui, et al. "Anomaly Transformer: Time Series Anomaly Detection with Association Discrepancy." International Conference on Learning Representations.
> [4] Tuli, Shreshth, Giuliano Casale, and Nicholas R. Jennings. "TranAD: deep transformer networks for anomaly detection in multivariate time series data." Proceedings of the VLDB Endowment 15.6 (2022): 1201-1214.
> [5] Wu, Haixu, et al. "TimesNet: Temporal 2D-Variation Modeling for General Time Series Analysis." The Eleventh International Conference on Learning Representations.
> [6] Chen, Yuhang, et al. "ImDiffusion: Imputed Diffusion Models for Multivariate Time Series Anomaly Detection." Proc. VLDB Endow. (2023).
> [7] Fan, Jin, et al. "LUAD: A lightweight unsupervised anomaly detection scheme for multivariate time series data." Neurocomputing 557 (2023): 126644.
> [8] Sugawara, Shota, and Ryuji Imamura. "PUAD: Frustratingly simple method for robust anomaly detection." 2024 IEEE International Conference on Image Processing (ICIP). IEEE, 2024.
> [9] Liu, Zhe, et al. "Multivariate time-series anomaly detection based on enhancing graph attention networks with topological analysis." Proceedings of the 33rd ACM International Conference on Information and Knowledge Management. 2024.

---

> ### Author Response · Authors · 2025-11-21
> **Response to Reviewer dAaG — Question #4, #6**
>
> ## Question \#4
> > The study mainly investigates multivariate time series but does not address the univariate case.
>
> We appreciate your suggestion. The study primarily focuses on the multivariate setting because most real-world monitoring environments naturally generate multivariate time series. Industrial and cyber-physical systems (CPS) typically operate with many sensors running simultaneously.
>
> This characteristic is well documented in prior work, which notes that modern CPS and industrial systems produce substantial amounts of multivariate time-series data [1, 2], and surveys consistently report that multivariate data is the dominant form across application domains such as manufacturing, healthcare, and finance [3]. The multivariate case is also more challenging and practically more relevant, as it requires capturing both temporal patterns and dependencies across variables. Recent studies indicate that multivariate time-series modeling is inherently more difficult due to the need to capture both temporal and cross-variable dependencies [4], and that anomaly detection becomes significantly more difficult in the presence of complex inter-variable relationships [5].
>
> For these reasons, our work focuses on multivariate anomaly detection, which better reflects real operational conditions.
>
> > Furthermore, it is unclear how model scalability behaves with respect to both sequence length and feature dimensionality, as no explicit analysis of these factors is provided.
>
> Regarding your question on scalability with respect to sequence length, we agree that this is an important consideration. However, varying the sequence length is not directly comparable across the entire set of models we study. Traditional methods treat each timestamp as an independent instance and do not operate on windowed sequences. As soon as the data are windowed, these methods cannot produce anomaly scores aligned with the original timeline. In contrast, deep learning models require fixed-length windows. Because the two families fundamentally differ in how sequence length is defined, changing the window length would prevent a fair comparison across all models. For this reason, sequence-length scalability was not evaluated.
>
> In contrast, scalability with respect to feature dimensionality can be evaluated fairly across all methods. Following your suggestion, we conducted an additional experiment varying the input dimensionality, and included the results in Section 4.3. Consistent with our main findings, we observe that the growth pattern of FLOPs with increasing dimensionality mirrors the behavior seen when increasing the number of instances.
>
> We thank you for this helpful suggestion and have incorporated the additional dimensionality analysis and clarifications into the revised manuscript.
>
> [1] Li, Dan, et al. "MAD-GAN: Multivariate anomaly detection for time series data with generative adversarial networks." International conference on artificial neural networks. Cham: Springer International Publishing, 2019.
> [2] Zhang, Yuxin, et al. "Unsupervised deep anomaly detection for multi-sensor time-series signals." IEEE Transactions on Knowledge and Data Engineering 35.2 (2021): 2118-2132.
> [3] Wang, Fengling, et al. "A survey of deep anomaly detection in multivariate time series: taxonomy, applications, and directions." Sensors (Basel, Switzerland) 25.1 (2025): 190.
> [4] Behrouz, Ali, Michele Santacatterina, and Ramin Zabih. "Chimera: Effectively modeling multivariate time series with 2-dimensional state space models." Advances in Neural Information Processing Systems 37 (2024): 119886-119918.
> [5] Han, Xiao, et al. "Root Cause Analysis of Anomalies in Multivariate Time Series through Granger Causal Discovery." The Thirteenth International Conference on Learning Representations. 2025.
>
> ## Question \#6
> The paper would benefit from a deeper analysis of when and why lightweight or heavy models perform better. For instance, cases where LSTM-VAE achieves relatively low FLOPs but competitive performance compared to heavier models like LOF need further investigation. Beyond aggregate AUROC scores across the entire dataset, it would be useful to analyze performance across different anomaly types and anomaly ratios to better contextualize the observed trends.
>
> As we mentioned in response to weakness \#1, the intention of our study is not to provide an exhaustive benchmark, but rather to investigate two focused research questions. Nonetheless, we appreciate you for suggesting a valuable direction for future work.

---

> ### Author Response · Authors · 2025-11-21
> **Response to Reviewer dAaG — Weakness #2 / Question #5**
>
> ## Weakness \#2
> > Limited theoretical or diagnostic interpretation of accuracy-runtime trade-off between lightweight and heavy models.
>
>  The goal of the study is to provide a diagnostic view of the trade-off between effectiveness and efficiency by examining how computational requirements scale across lightweight and heavy models. Rather than offering theory in the sense of formal learning guarantees, our analysis relies on FLOPs as a principled and model-agnostic theoretical measure of algorithmic complexity.
>
>  FLOPs provide a direct representation of the computation required by each method and therefore serve as a theoretical lens for interpreting why heavy models incur substantially higher complexity costs yet do not consistently yield superior detection performance in practice. This FLOPs-based analytical framework is what enables us to reveal the fundamental mismatch observed between computational cost and accuracy.
>
>  We hope you will acknowledge our effort to establish a fair and hardware-agnostic basis for comparison. In particular, we derived closed-form FLOPs expressions for all traditional models, which has not been addressed in prior works, allowing traditional machine learning and deep learning methods to be evaluated under a unified computational metric.
>
> ## Question \#5
> > While adopting FLOPs as a hardware-agnostic proxy is a good practice, complementing it with real runtime measurements and memory footprint analyses across identical hardware would provide stronger evidence for practical applicability in real deployments.
>
> As requested, we have complemented the FLOPs-based analysis with real runtime measurements to strengthen the practical relevance of our findings. In Section 4.2 of the revised manuscript, we report training and inference times measured on our highly resourced environment (with GPU acceleration disabled to ensure a fair CPU-only comparison).
>
> Using the SMD dataset and selecting representative methods from each category, we observed that traditional models such as HBOS and Isolation Forest complete both training and inference within a few seconds, whereas deep learning approaches incur significantly higher costs. In particular, ABOD and USAD required hundreds to thousands of seconds. The table below summarizes the measured runtimes:
>
> | model   | train time | inference time |
> |---------|-----------:|---------------:|
> | ABOD    | 4700.05    | 6331.65        |
> | HBOS    | 3.27       | 0.77           |
> | IForest | 4.97       | 2.01           |
> | USAD    | 1047.60    | 5.46           |
>
> We believe these measurements closely align with the FLOPs-based trends and provide concrete evidence supporting the practical infeasibility of computationally heavy models in real-world, resource-constrained deployments.

---

### Official Review · Reviewer_YBTB · 2025-11-01

**Soundness:** 3
**Presentation:** 2
**Contribution:** 2
**Rating:** 4
**Confidence:** 4

**Summary:**

This paper presents an empirical study challenging the necessity of "heavy" deep learning models for multivariate time series anomaly detection (MTS-AD), particularly in resource-constrained environments . The authors conduct a comparative analysis of statistical, classical machine learning (one-class), and deep learning (reconstruction-based) methods across six benchmarks. The study evaluates models on two axes: detection effectiveness (using the threshold-agnostic AUROC metric) and computational efficiency (using a hardware-agnostic FLOPs metric).

The study finds that traditional models (e.g., HBOS, Isolation Forest, ABOD) frequently achieve top-tier AUROC performance, often matching or exceeding their deep learning counterparts . Furthermore, the analysis of AUROC vs. FLOPs (Figure 1) suggests that deep learning models present a poor trade-off, incurring high computational costs without delivering superior performance. The authors conclude that deep learning is not uniformly superior and that lightweight traditional models are often the more practical choice for constrained deployments .

**Strengths:**

1. The paper asks a timely and important question: are heavy deep models worth the cost for MTS-AD in real-world, constrained settings ? This is a critical concern for practitioners in industrial, IoT, and embedded systems.
2. The joint evaluation of both detection performance (AUROC) and hardware-agnostic computational cost (FLOPs) is a valuable contribution.

**Weaknesses:**

1. The paper's primary flaw lies in its selection of deep learning models. The entire "Reconstruction" (i.e., deep learning) category consists almost exclusively of older, simpler autoencoder variants (LSTM-AE, LSTM-VAE, USAD, DeepSVDD). These models are no longer representative of the state-of-the-art. The study omits the entire class of modern, high-performance Transformer-based and CNN-based anomaly detectors. The comparison is therefore not "Rethinking Heavy Models" but "Rethinking Outdated Models".
2. The paper incorrectly frames "deep learning" as "heavy" and "traditional" as "lightweight," when its own data often shows the opposite. This invalidates the core narrative.
3. While AUROC is threshold-agnostic, it is notoriously unreliable for anomaly detection on datasets with high class imbalance (which is characteristic of all TSAD benchmarks) . AUPRC (Area Under Precision-Recall Curve) is the standard, more informative metric in this setting. By optimizing for a potentially misleading metric (AUROC), the performance rankings (Table 3) may not reflect true detection quality. This choice further weakens the paper's conclusions.

**Questions:**

Re: Weakness #1: The paper's conclusions hinge on comparing deep learning to traditional methods. Why did the authors choose to represent the entire deep learning category with only older, reconstruction-based models, while omitting all modern SOTA architectures like Anomaly Transformer, TimesNet, or TranAD, which are the "heavy models" the community is actually discussing today?

Re: Weakness #2: The FLOPs data in Table 3 (e.g., SMD, SMAP) shows that statistical models like ABOD and LOF are orders of magnitude more computationally expensive (higher FLOPs) than deep models like OmniAnomaly or LSTM-AE. How do the authors reconcile this fact with the paper's central narrative that deep learning models are the "heavy" option and traditional models are the "lightweight" alternative?


Re: Weakness #3: Why did the authors choose AUROC as the sole metric for detection performance, given that AUPRC is widely accepted as a far more informative and reliable metric for highly imbalanced anomaly detection tasks?

---

> ### Author Response · Authors · 2025-11-21
> **Response to Reviewer YBTB — Weakness #1 / Quenstion #1**
>
> Thank you for pointing out these important questions. We respond to each weakness-question pair below.
>
>
> ## Weakness #1 - Question #1
> >  Why did the authors choose to represent the entire deep learning category with only older, reconstruction-based models, while omitting all modern SOTA architectures like Anomaly Transformer, TimesNet, or TranAD, which are the "heavy models" the community is actually discussing today?
>
>
> Our primary objective was to examine the performance–computational cost trade-off under realistic deployment constraints. In forming the deep learning group, we prioritized methods that satisfy the criteria outlined in Section 3.2.1:
> (1) chronological coverage
> (2) diverse methodological mechanisms
> (3) widespread adoption (highly cited) as baselines
> Models such as LSTM-AE, LSTM-VAE, USAD, DeepSVDD, and OmniAnomaly meet these criteria and continue to be part of mainstream benchmark suites.
> Importantly, we initially excluded extremely heavy modern architectures (e.g. Transformer-based and large CNN-based models) because their computational profile is already well-known to be significantly higher than traditional methods and classical deep learning baselines. Given their architectural complexity and parameter counts, such models are a priori expected to incur much larger FLOPs, making them clear outliers within the compute spectrum we aimed to analyze.
>
> That said, we agree with you that including representative modern architectures strengthens the paper, especially given the community’s increasing focus on heavy Transformer/CNN-based anomaly detectors. We therefore incorporated **Anomaly Transformer and TimesNet into the main paper**, both of which are widely recognized as computationally demanding state-of-the-art models.
>
> As expected, these models exhibit substantially higher computational cost than most of the baselines, yet their AUROC does not consistently surpass lightweight statistical models. Their inclusion thus reinforces the core claim of our paper: that heavier modern deep models do not guarantee superior performance, while their compute demands make them difficult to deploy in resource-constrained environments.
> Thank you for this valuable suggestion. We have revised the paper accordingly and clarified our model selection rationale.

---

> ### Author Response · Authors · 2025-11-21
> **Response to Reviewer YBTB — Weakness #2 / Quenstion #2**
>
> ## Weakness #2 - Question #2
> > How do the authors reconcile this fact with the paper's central narrative that deep learning models are the "heavy" option and traditional models are the "lightweight" alternative?
>
> We appreciate you for pointing out this important issue. We agree that our initial framing may have implied an overly binary distinction between “traditional = lightweight” and “deep learning = heavy,” and we acknowledge that this does not universally hold. As our own results indicate, k-NN-based traditional methods such as ABOD and LOF can be extremely costly, in some cases exceeding the FLOPs of certain deep models.
> Our intended message was not that all traditional models are lightweight or all deep models are heavy, but that many widely used traditional detectors tend to be computationally lighter, while modern deep architectures generally require substantially more operations. The k-NN family is a well-known exception, and Section 4.3 already discusses their dramatic computational growth.
> However , you are right that this nuance needed to be stated more clearly. In response, we have revised the manuscript to sharpen our claim:
> - We directly identify ABOD and LOF as traditional but heavy methods repeatedly in Section 4, across Section 4.1, 4.2, and 4.3.
> - We have revised Section 5 to adopt a more balanced framing and avoid binary terminology.
>
> We thank you again for prompting this clarification, which has improved the accuracy and precision of our narrative.

---

> ### Author Response · Authors · 2025-11-21
> **Response to Reviewer YBTB — Weakness #3 / Quenstion #3**
>
> ## Weakness #3 - Question #3
> > Why did the authors choose AUROC as the sole metric for detection performance, given that AUPRC is widely accepted as a far more informative and reliable metric for highly imbalanced anomaly detection tasks?
>
> Our choice of AUROC as the primary evaluation metric was motivated by the need for a fair and consistent comparison across datasets. Since AUROC is largely invariant to class imbalance, it provides a stable measure of ranking quality even when anomaly ratios differ substantially. This was particularly important in our setting, as the six datasets we evaluate span a wide range of anomaly ratios (4.16%–27.76%). In contrast, AUPRC is highly sensitive to the anomaly ratio, which would make cross-dataset comparisons difficult, as lower anomaly ratios naturally yield lower AUPRC values regardless of the underlying model behavior. This strong dependence of AUPRC on the anomaly ratio can be clearly observed in Appendix E.1 Figure 4.
>
> Also, we want to state that our decision aligns with existing well-known anomaly detection benchmarks, [1] and [2]. They adopt AUROC as their main comparison metric, reflecting its role as a standard and threshold-free evaluation measure.
>
> At the same time, we fully agree that AUPRC offers a more sensitive view within individual datasets, where the anomaly ratio is fixed and the metric more directly captures a model’s ability to identify rare events. Following your helpful suggestion, we have incorporated AUPRC into all main performance tables and detecion performance-efficency trade-off visualizations. Additionally, we now report VUS-ROC and VUS-PR [3] in the appendix to provide broader perspectives on model performance.
>
> [1] Han, Songqiao, et al. "Adbench: Anomaly detection benchmark." Advances in neural information processing systems 35 (2022): 32142-32159.
> [2] Liu, Qinghua, and John Paparrizos. "The elephant in the room: Towards a reliable time-series anomaly detection benchmark." Advances in Neural Information Processing Systems 37 (2024): 108231-108261.
> [3] Paparrizos, John, et al. "Volume under the surface: a new accuracy evaluation measure for time-series anomaly detection." Proceedings of the VLDB Endowment 15.11 (2022): 2774-2787.

---

### Meta-Review · Area_Chair_Vhtx · 2026-01-04

**Summary:**

There are several concerns, especially concentrating on appropriate evaluation measures for this task, limited datasets, and limited compared methods. The authors made an effort to address these comments; however, not fully satisfactory. For example, VUS-based measures were presented but only in appendix while main paper still focuses on older measures. Similarly, baselines and datasets used do not seem sufficient. Reviewers suggested following recent benchmark practices (e.g., TSB-AD, VUS) and heavily focus the analysis around these datasets/measures. I hope the authors will find the comments useful, update the manuscript, and find an alternative place for this work in the future.

**Reviewer Concerns:**

THe authors now report partially results using VUS-based measures but still haven't used human-curated datasets that eliminated biases (e.g., TSB-AD) or stronger baselines as reported in such recent benchmarks.

**Reviewer Scores:**

Scores are below bar for acceptence and even if we assume all reviewers would raise their scores, the paper is still uncertain whether it would be accepted.

---

### Decision · Program_Chairs · 2026-01-26

Reject